# Crystal-facet-dependent surface transformation dictates the oxygen evolution reaction activity in lanthanum nickelate

Achim Füngerlings [1], Marcus Wohlgemuth[2], Denis Antipin[3], Emma van der Minne [4], Ellen Marijn Kiens [4], Javier Villalobos [3], Marcel Risch [3], Felix Gunkel [2], Rossitza Pentcheva [1] ✉ & Christoph Baeumer [4] ✉

Electrocatalysts are the cornerstone in the transition to sustainable energy technologies and chemical processes. Surface transformations under operation conditions dictate the activity and stability. However, the dependence of the surface structure and transformation on the exposed crystallographic facet remains elusive, impeding rational catalyst design. We investigate the (001), (110) and (111) facets of a $LaNiO_{3-\delta}$ electrocatalyst for water oxidation using electrochemical measurements, X-ray spectroscopy, and density functional theory calculations with a Hubbard $U$ term. We reveal that the (111) overpotential is $\approx 30-60$ mV lower than for the other facets. While a surface transformation into oxyhydroxide-like NiOO(H) may occur for all three orientations, it is more pronounced for (111). A structural mismatch of the transformed layer with the underlying perovskite for (001) and (110) influences the ratio of $Ni^{2+}$ and $Ni^{3+}$ to $Ni^{4+}$ sites during the reaction and thereby the binding energy of reaction intermediates, resulting in the distinct catalytic activities of the transformed facets.

Improving the catalytic activity and stability in electrochemical energy conversion processes - a decade-old endeavor in an interdisciplinary research field - gains more and more urgency due to our need for defossilisation of energy and industrial processes. Storing electrical energy in the $H_2$ chemical bond via electrochemical water splitting is a key example. Perovskite-type oxides ($ABO_3$) are attractive catalysts for the oxygen evolution reaction (OER), the kinetically limiting half-cell reaction during water electrolysis. Their tunability allows optimization of OER activity according to electronic structure descriptors like the electron occupancy of frontier orbitals[1], relative orbital position[2], and the bond covalency between the transition metal ion and oxygen[3,4]. These electronic properties determine the binding energies of the

reaction intermediates on the catalyst surface, which are decisive for the reactivity. Yet, most catalyst synthesis routes in research and application inherently lead to rough and ill-defined surface morphologies, exposing an ensemble of different crystallographic facets. Advancing our understanding on the role of surface orientation and termination on the catalytic activity[5–11] is therefore essential for the rational design of catalysts.

This situation is additionally complicated by the fact that many catalyst surfaces undergo restructuring under reaction conditions in diverse chemical environments[12–21]. For example, we have shown experimentally that the (001) Ni-terminated $LaNiO_{3-\delta}$ perovskite surface (one of the most active ternary oxide OER electrocatalysts)

[1]Department of Physics, Theoretical Physics and Center for Nanointegration (CENIDE), University of Duisburg-Essen, Lotharstraße 1, Duisburg 47057, Germany. [2]Peter Gruenberg Institute and JARA-FIT, Forschungszentrum Juelich GmbH, Juelich, Wilhelm-Johnen-Straße, Jülich 52428, Germany. [3]Nachwuchsgruppe Gestaltung des Sauerstoffentwicklungsmechanismus, Helmholtz-Zentrum Berlin für Materialien und Energie GmbH, Hahn-Meitner-Platz 1, Berlin 14109, Germany. [4]MESA+ Institute for Nanotechnology, Faculty of Science and Technology, University of Twente, Hallenweg 15, Enschede 7522, Netherlands. ✉e-mail: rossitza.pentcheva@uni-due.de; c.baeumer@utwente.nl

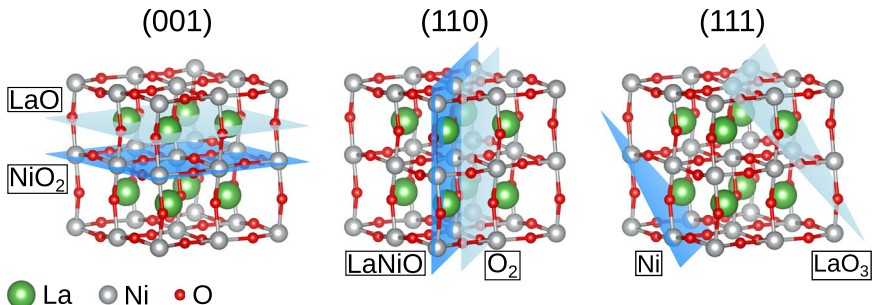

**Fig. 1 | Facet-dependent surface terminations of LaNiO₃.** The LaNiO₃ perovskite structure with lattice planes indicating the two different terminations for each of the three investigated crystallographic orientations (for better visibility, the highlighted planes for (111) are not adjacent).

undergoes a transformation to an oxyhydroxide-like structure before the onset of OER, which was further substantiated by the presence of a redox peak before OER onset. Moreover, the experimental OER activity trends of the (001) LaNiO$_{3-\delta}$ surface can only be rationalized if the surface phase transformation is taken into account in the density functional theory with Hubbard $U$ (DFT+$U$) modeling[12]. Similar surface transformations are found for Co-containing electrocatalysts[14,17], which may be connected to the frequently observed inverse activity-stability relation[22,23]. Phenomenologically, different crystal facets were reported to yield different activity and stability. However, any deeper understanding of the role of facet-dependent surface-transformations of highly dynamic surfaces like LaNiO$_{3-\delta}$ and the resulting surface-structure under reaction conditions remains elusive. This impedes targeted synthesis of LaNiO$_{3-\delta}$-based electrocatalysts with optimized crystallographic properties and - more generally - limits our understanding of facet-dependent transformations and resulting activity trends of the electrocatalyst surface.

In this work, we investigate the influence of surface orientation on the surface transformation and the OER activity of (001), (110) and (111)-oriented LaNiO$_{3-\delta}$ electrocatalysts (Fig. 1). Experimentally, we investigate epitaxial perovskite thin films, which are ideally suited for the comparison of different crystallographic facets, in contrast to typically employed particulate-based electrocatalysts[5,8,22,24–26]. These thin films enable a one-to-one comparison to single facets modelled by DFT[7,9,11,27–29] and they allow tracing the evolution of the surface phase with up to unit-cell-thickness sensitivity[12,17]. Based on DFT+$U$ calculations, we compare the surface chemistry and OER activity of the three LaNiO$_{3-\delta}$ facets. Our experimental and computational results suggest that the highly OER-active, Ni-terminated (111) facet undergoes a similar surface phase transformation like the previously investigated, less active (001) facet. The detailed DFT+$U$ analysis of the underlying properties reveals intricate orientation dependence of the thickness, structural and electronic properties of the emerging oxyhydroxide-like NiOO surface layer with edge-sharing octahedra. Due to the structural mismatch to the underyling perovskite layer and modified ratio between Ni$^{3+}$ and Ni$^{4+}$, the oxygen intermediate is less strongly bound on the transformed (001) facet, which leads to a high overpotential for the potential determining *OH→*O step. In contrast, the structural match between the oxyhydroxide-like surface layer and the (111) perovskite facet leads to enhanced OER activity and close to equal reaction free energy steps of intermediates approaching the ideal characteristics.

## Results
### Experimental determination of facet-dependent activity
We synthesized epitaxial 20 nm LaNiO$_{3-\delta}$ thin films on SrTiO₃ substrates with (001), (110) and (111) orientation (using the pseudocubic description of the unit cell), see methods for details. In situ reflection high energy electron diffraction (RHEED) reveals a two-dimensional growth mode (Fig. 2a and Supplementary Fig. 1). X-ray diffraction

(XRD) confirms that the films are grown fully epitaxially, exposing the desired facets in each case (Fig. 2b). The (002), (110) and (111) film peaks show pronounced Laue fringes, indicative of coherent thickness and smooth interfaces for all three orientations (Fig. 2b). Atomic force microscopy (AFM) reveals smooth, step-terraced surface morphologies, as shown in Fig. 2c. Scanning transmission electron microscopy (STEM) analysis overall confirms the epitaxial growth (Supplementary Fig. 2), while revealing a number of extended defects such as dislocations and antiphase boundaries. These are expected for the chosen synthesis temperature, which was selected to achieve predominant Ni-termination[12]. X-ray photoelectron spectroscopy (XPS) investigation after in situ transfer from the PLD to the analysis chamber (Supplementary Fig. 3) reveals relatively higher Ni 3$p$ intensities for smaller mean escape depths $d$ for LaNiO$_{3-\delta}$ (001) and (111) thin films, after normalization to the La 4$d$ peaks. This observation suggests that the (001) and (111) thin films possess the desired predominant Ni-termination, corresponding to the NiO₂ and Ni planes shown in Fig. 1 (see refs. 12,30 for detailed discussion of termination-dependent XPS intensities). For LaNiO$_{3-\delta}$ (110), all cation-containing planes possess equal amounts of La and Ni, and we observe similar Ni 3$p$ and La 4$d$ intensities for both mean escape depths.

The Ni-rich termination in the (001) and (111) facets is also reflected in the electrochemical response (Fig. 3a), where the Ni termination leads to pronounced redox waves at a potential of 1.4 V versus the reversible hydrogen electrode (RHE). Previously, we found that this redox wave corresponds to a surface phase transformation to a oxyhydroxide-like layer, which is consistent with the well-known redox feature in the Ni(OH)₂/NiOOH system[12]. Interestingly, the redox wave is more pronounced for (111) facets compared to (001), which might indicate a more pronounced phase transformation. The area underneath the redox peaks was integrated to measure the charge associated with the Ni redox, a well-established method to quantify the number of electrochemically accessible metal cations[31]. The inset of Fig. 3a shows the measured charge in cyclic voltammograms recorded before and after exposing the sample to OER conditions (see methods for details). We compare this value to the charge expected for redox involving all Ni ions in a single atomic layer on the perovskite surfaces, which depends on the orientation. After the first OER cycles, the measured charge is similar to the expected charge of one Ni layer for (001). Yet for (111), the measured charge is twice as high as expected for a single layer even in the first cycle, indicating that more than one Ni layer participates in the redox phenomenon for the (111) facet, a point to which we will return below.

We analyzed the electrocatalytic OER activity for the (001), (110) and (111) LaNiO$_{3-\delta}$ layers using cyclic voltammetry (Fig. 3c, d) and (staircase) chronopotentiometry (Fig. 3e, f). Chronopotentiometry minimizes the influence of additional currents (compared to typical cyclic voltammetry activity testing) unrelated to the OER such as capacitive charging or oxygen intercalation into the perovskite lattice[32]. All samples were *iR*-corrected with the uncompensated

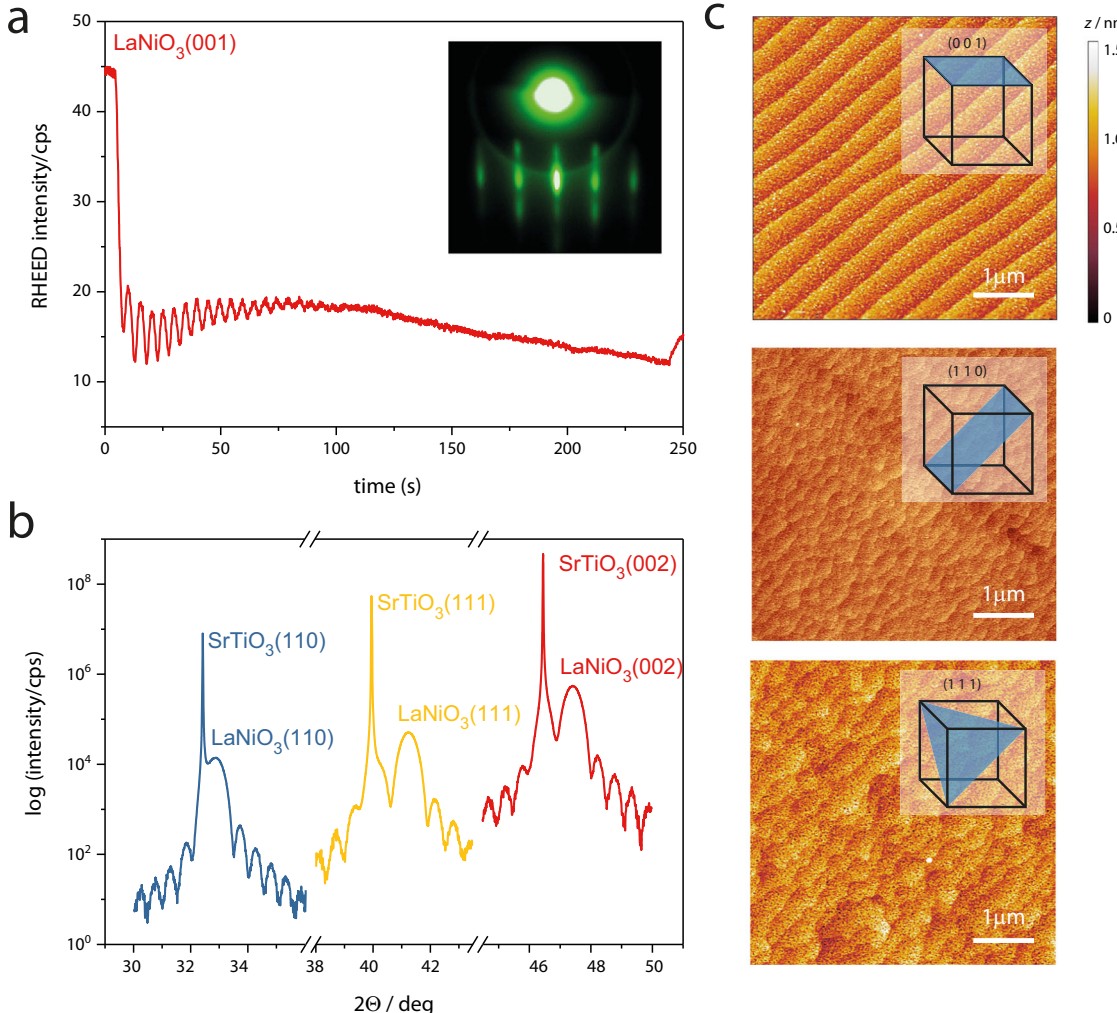

**Fig. 2 | Thin film characterization. a** RHEED intensity evolution during PLD growth of LaNiO$_{3-\delta}$ on a (001)-oriented SrTiO$_3$ substrate. Inset: RHEED pattern after growth. **b** X-ray diffractogram of representative 20 nm LaNiO$_{3-\delta}$ films in (001), (110) and (111) orientation. Each diffractogram shows the most intense substrate and film peaks for the given orientation. **c** AFM scans of the same films. The root-mean-square roughness is below 350 pm for all films.

resistance ($R_u$) obtained from the *x*-axis offset in Nyquist impedance plots as shown in Fig. 3b. Because of the low roughness, we approximate the actual oxide surface area with the geometric area. Double-layer capacitance measurements of the different facets confirm differences within <1.1%. Therefore, we compare current densities normalized to the geometric area. All LaNiO$_{3-\delta}$ films exhibited appreciable OER activities. The overpotential $\eta$ at a current density of 1 mA cm$^{-2}$ (the typical reference current density for epitaxial thin films) was 0.51 V, 0.49 V and 0.45 V for (001), (110) and (111) films, respectively, as shown in Fig. 3d, and in line with the decreasing Tafel slopes of 68.4, 60.9 and 50.4 mV/dec (Fig. 3f), respectively. This means that (111) samples are noticeably more active than the (001) and (110) counterparts, similar to the trend observed for SrRuO$_3$[26], but in contrast to Co/Fe-based perovskite oxide OER electrocatalysts, for which the (111) facet was found to be the least active[5,8,25]. The activity trend is even more pronounced when comparing the turn-over frequency (TOF) obtained by normalization of the current density to the number of Ni ions in a perfect perovskite surface (Supplementary Fig. 4). Interestingly, we also found the lifetime of the LaNiO$_{3-\delta}$ (111) facet increased compared to the (110) and (001) orientations, as we reported in reference[22], where we phenomenologically compared the lifetimes of differently oriented LaNiO$_{3-\delta}$ thin films via chronopotentiometry. Thus, the LaNiO$_{3-\delta}$ (111) facet is

both the most active and the most stable among our investigated facets, which means the often observed inverse activity-stability relation is overcome for this material system[22]. Summarizing, we find that the electrochemical behavior of the (111) facet is the most intriguing, because of the highest observed activity and stability, and the most pronounced Ni redox features. These observations may be indicative of a complex atomic-level surface structure and composition and raise the question how these surface properties differ from the surface transformation on the (001) facet, which we investigated in more detail before[12].

**Structural characterization**

To track changes in the catalyst chemical and electronic structure for (111) LaNiO$_{3-\delta}$, we performed X-ray absorption spectroscopy (XAS) before and after OER operation. We note that XAS is a bulk-sensitive characterization technique in the used measurement geometry, where the spectral contribution of a thin surface layer (≤1 nm) is about 5% as compared to the signal from the bulk (≈19–20 nm). The X-ray absorption near edge structure (XANES) and the extended X-ray absorption fine structure (EXAFS) of the Ni-K edge were collected on a pristine (111) LaNiO$_{3-\delta}$ film, and after chronoamperometry (CA) at 1.75 V vs RHE for 1 h (initial current density of 3.2 mA cm$^{-2}$) and after CA at 2.5 V vs RHE for 16 h (Fig. 4a). The latter represents an extreme

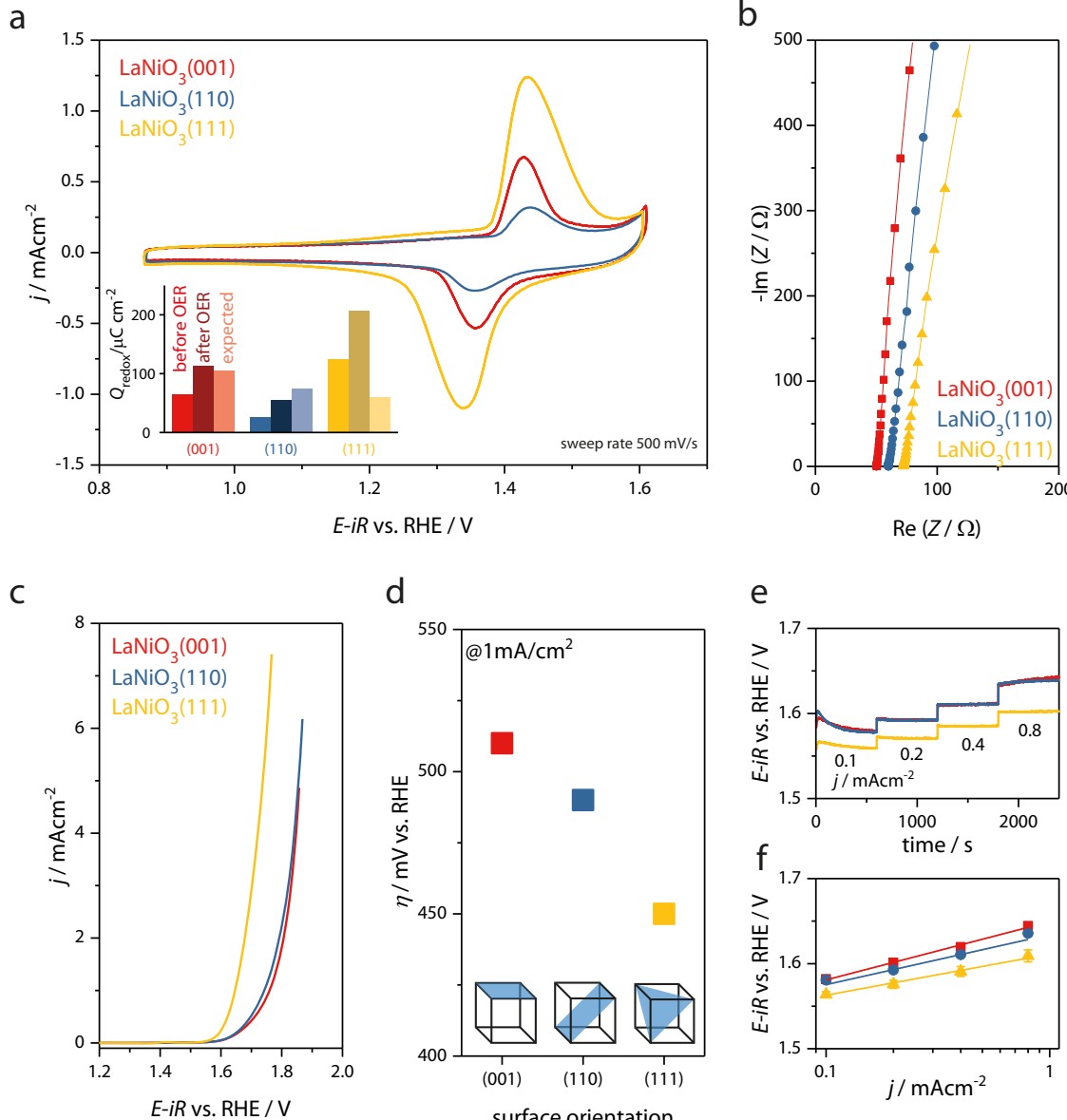

**Fig. 3 | Electrochemical performance. a** Cyclic voltammetry in the Ni redox region before applying OER potentials with a scan rate of 500 mV s⁻¹. Inset: Measured charge in cyclic voltammograms recorded before and after exposing the sample to OER conditions, compared to the charge expected for redox involving all Ni ions in a single atomic layer. **b** Electrochemical impedance spectroscopy, Nyquist plot of the high-frequency region to determine the uncompensated resistance for representative films in each orientation. **c** Cyclic voltammetry in the OER potential regime, showing the average between anodic and cathodic sweep of the second cycle. **d** Comparison of the experimental overpotential for a current density of 1 mA cm⁻² for the different surface orientations. **e** Steady-state measurement of four consecutive galvanostatic holds. **f** Tafel plot obtained from steady-state galvanostatic measurements. The potential values of the last minute in each 10 min step were averaged to calculate one potential value for each current density step, error bars represent the absolute variance for every data point.

condition, magnifying the expected changes and is used as a reference. The XANES of the $LaNiO_{3-\delta}$ films differs from that of the $Ni^{2+}O$ reference and is similar to that of $LiNi^{3+}O_2$, particularly in the rising of the spectrum between 8335 and 8348 eV. On closer inspection, $LiNi^{3+}O_2$ has a shoulder at 8343 eV that is due to metal 4p states in the edge-sharing octahedra of ordered transition metal oxides.[33] This shoulder is absent in the as-prepared $LaNiO_{3-\delta}$.

In contrast, the $LaNiO_{3-\delta}$ sample after 16 h CA at 2.5 V vs. RHE shows clear changes in the rising part at lower energy and a broader maximum. The shift of the edge rise to lower energy suggests a reduced Ni valence[34] and/or the appearance of the aforementioned shoulder due to a transformation from corner-sharing (perovskite) to edge-sharing octahedra (e.g., layered hydroxide or rocksalt). The broader maximum, the so-called white line, suggests a less ordered structure and/or a mix of more than one crystal phase. The $LaNiO_{3-\delta}$ sample after 1 h CA at 1.75 V is almost identical to the pristine one except for a minute shift of the rising part to lower energy, which may be a result of a similar transformation as observed after 16 h, albeit to a much smaller extent. As small changes on the surface are expected, we calculated differential XANES spectra ($\Delta\mu$ method)[35]. We plot the intensity difference between treated samples (1h/1.75V and 16h/2.5V) and as-grown sample in Fig. 4b. The qualitative similarity in the difference spectra supports that the changes clearly visible in the sample after 16 h at 2.5 V were also present on the surface of the sample after 1 h at 1.75 V. The quantitatively small differences observed for the sample after 1 h at 1.75 V are expected for a chemical transformation of the surface-near atomic layers, given the extended information depth of fluorescence-mode XAS.

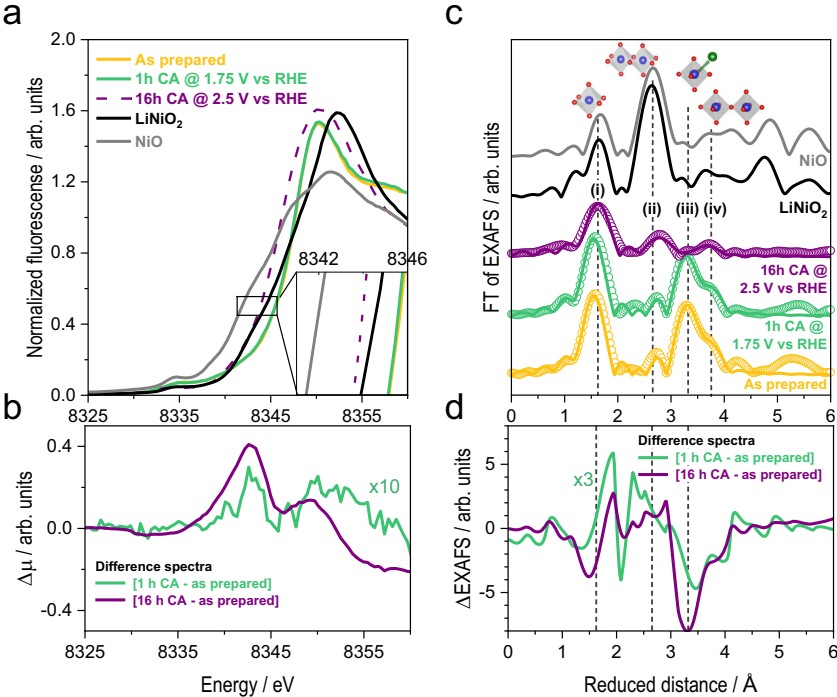

**Fig. 4 | X-ray absorption analysis. a** XANES spectra of Ni-K edge collected on (111)-oriented LaNiO$_{3-\delta}$ samples (as prepared, after 1h and 16h potential hold). The Ni-K edge spectra of Ni$^{2+}$O and LiNi$^{3+}$O$_2$ were added as references. **b** $\Delta\mu$ spectra obtained by subtracting the as-prepared spectrum from the one after 16 h CA and 1 h CA, the latter multiplied by a factor of 10, showing similarity of spectral changes. **c** Fourier transform of the EXAFS of the Ni-K edge collected on (111)-oriented LaNiO$_{3-\delta}$ samples. Data are shown as circles and EXAFS fits as lines. Fourier transforms were performed between 35 and 550 eV (above E$_0$ = 8333 eV) with a cosine window covering the first and last 10% of the data range. Interatomic distances and fitting parameters are listed in Supplementary Tables 1, 2, 3, 4. **d** $\Delta$EXAFS of Fourier-filtered data (0.0 - 4.5 Å) obtained by subtracting the as-prepared spectrum from the one after 16 h CA and 1 h CA, the latter multiplied by a factor of 3, showing similarity in spectral changes.

To gain further insight into the structure and Ni valence, we analyzed the Fourier transform (FT) of the EXAFS at the Ni-K edge of (111)-oriented LaNiO$_{3-\delta}$ (Fig. 4c) and performed EXAFS fits. A detailed discussion of the EXAFS fits and the extracted interatomic distance, $R$, is provided in Supplementary Note 2, Supplementary Figs. 5, 6 and Supplementary Tables 1–7. The pristine LaNiO$_{3-\delta}$ and LaNiO$_{3-\delta}$ after 1 h CA have a nearly identical EXAFS (Fig. 4c) and thus nearly identical structure. The most intense peaks (i) and (iii) are assigned to Ni-O and Ni-La, respectively, indicative of the perovskite structure[36,37]. For the (001) surface after 1 h CA at 1.75 V vs. RHE, the EXAFS remains similar to the pristine state, akin to the (111) surface (Supplementary Fig. 6). We note that EXAFS peaks indicate the bond distances of highest occurrence[34]. The data thus indicate that the films mostly remained in perovskite phase, in line with our XRD, XPS and STEM characterization (Supplementary Figs. 7–9). In Supplementary Note 3, we extensively discuss how the additional structural characterization confirms that only a thin surface layer transforms during operation at 1.75 V vs RHE.

The EXAFS of LaNiO$_{3-\delta}$ after 16 h at 2.5 V is markedly different. The Ni-O peak (i) is found at larger distance, indicative of a reduction of Ni, possibly related to a reduced hydroxide or oxide, e.g., Ni(OH)$_2$ or NiO. This is expected in our ex-situ measurement, where the samples were returned to open circuit voltage (OCV) and exposed to air before X-ray absorption analysis. Moreover, the (111) sample operated for 16 h at 2.5 V exhibited no clear Ni-La peak (iii) and instead a new peak (ii) appeared, which is not expected for the perovskite structure. Similar observations were made previously for Co-based perovskites, where the peak was assigned to a metal-metal distance in edge-sharing octahedra[19], which is also supported by the similar position of peak (ii) to the edge-sharing reference LiNiO$_2$ (Fig. 4c). Thus, we assigned the new peak to edge-sharing Ni octahedra (Supplementary Fig. 5b,c) and note that the corner-sharing Ni octahedra, i.e., peak (iv), are resolved simultaneously. We again calculated differential spectra ($\Delta$EXAFS), where we observe qualitatively similar features for both treated samples. For the sample after 1 h at 1.75 V, only small changes are expected because the transformation is confined to the surface. The difference spectra thus support that the changes visible after 16 h at 2.5 V were also present on the surface of the sample after 1 h at 1.75 V (Fig. 4d), again implying a hydroxide-like surface phase.

In summary, XANES and EXAFS support that applying 1.75 V vs. RHE results in the formation of a new disordered (surface) phase with edge-sharing octahedral symmetry, which may be related to the previously observed nickel-oxyhydroxide-type NiOO surface layer. After returning to OCV, the Ni valence of this edge-sharing surface phase is reduced compared to the pristine LaNiO$_{3-\delta}$ film. All in all, the (111) facet apparently undergoes a similar surface phase transformation towards a hydroxide-like NiOO layer with edge-sharing octahedra as observed for the (001) facet[12], but the details of the surface structure, valence states and coordination chemistry may vary between both facets. We note that due to the low solubility of La(OH)$_3$, the La species remain on the surface[17], as confirmed by XPS analysis after 16 h operation, where the La to Ni ratio did not change appreciably (Supplementary Fig. 7).

**Theoretical insight into the origin of facet-dependent activity**

To gain further insight into the facet-dependent OER activity and the structural and electronic features of the transformed LaNiO$_3$ surfaces, we carried out a detailed DFT+$U$ investigation for the three facets, (001), (110) and (111), both transformed and untransformed. In the transformed surfaces, the lateral density of Ni in the edge-sharing NiOO(H) layer is higher than in the perovskite layers. Consequently, more than one Ni-containing perovskite layer is involved in the formation of one layer of NiOO(H), raising the question which perovskite termination is present at the interface to the NiOO(H) layer. For the

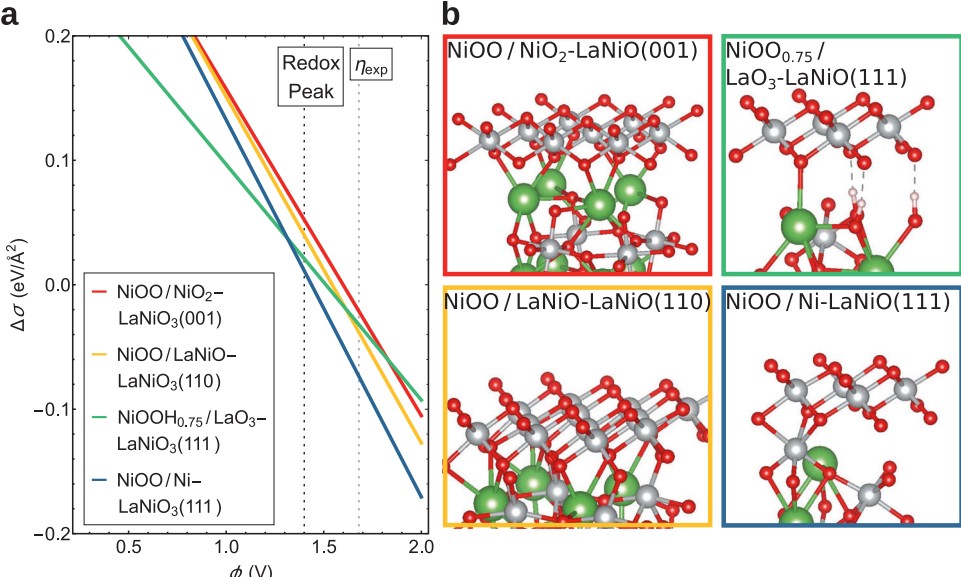

**Fig. 5 | NiOOH$_x$/LaNiO$_3$ interface stability from DFT+$U$. a** Relative stability of NiOOH$_x$ on (001), (110) and (111)-oriented perovskites as a function of applied voltage at pH = 13. Only the most stable systems with respect to the number of intercalated hydrogen are shown for each orientation (see side views of the structures in **b**, for an overview over all systems, see Supplementary Fig. 10).

(001) facet, it was established that NiOO on a LaO-termination is more stable than on NiO$_2$[12]. We compared the interface binding energy of NiOOH$_x$, where $x$ denotes the amount of hydrogen intercalated at the interface, to both possible terminations of the (110) and (111) facets, as well as the LaO-terminated (001) surface. As shown in Fig. 5, for the (111) oriented surfaces, the two surface structures, NiOOH$_{0.75}$ on LaO$_3$-terminated LaNiO$_3$(111) and NiOO on Ni-terminated LaNiO$_3$(111), compete in energy in the vicinity of the redox peak. Therefore, both were considered in the further analysis. Fig. 5a also indicates that the (111)-oriented transformed layers are more stable than the ones at the (001) and (110) facets. Owing to the higher amount of oxygen anions at the LaO$_3$ interface, H-intercalation of up to 3 H per unit cell is energetically favorable only for the LaO$_3$-terminated LaNiO$_3$(111) and leads to an ordered arrangement, resembling a bulk oxyhydroxide. Since the cyclic voltammetry data of the (111) facet indicates a higher thickness of the transformed layer, even in the first cycle, the models for the transformed (111) surface shown in Fig. 5b were extended by an additional NiOOH layer, i.e. NiOOH/NiOOH$_{0.75}$/LaO$_3$-LaNiO$_3$(111) and NiOOH/NiOO/Ni-LaNiO$_3$(111).

Prior to investigating the OER activity, we considered the most stable coverage with functional groups under reaction conditions for each facet in a surface Poubaix diagram shown in Supplementary Figs. 11 and 12[38]. For the untransformed (001) NiO$_2$-termination, it was shown previously that all Ni sites are hydroxylated and a quarter of the O sites are hydrogenated under reaction conditions[12]. We find that the untransformed LaNiO-terminated (110)-surface is covered by one-quarter of oxygen and three-quarters of $^*$OOH. The untransformed (111) Ni-termination is covered either by a full monolayer (ML) $^*$OOH or by 3/4 ML $^*$OH + 1/4 ML $^*$OOH. On the other hand, the transformed NiOO surface layer on LaNiO$_3$(111) was found to be fully oxygen-terminated under reaction conditions, similar to previous findings for NiOO on the (001) facet[12]. In contrast, NiOO on LaNiO$_3$(110) exhibits a mixture of two-thirds oxygen- and one-third hydroxyl-coverage.

We now turn to a comparison of the calculated overpotentials to the experimental ones, noting that the latter depend on the chosen current density and include reaction kinetics, while the DFT-derived overpotentials describe the thermodynamic potential barrier. As shown in Fig. 6a, for the untransformed (001) NiO$_2$-facet, we obtain an overpotential of 0.87 V, for both a Ni and a lattice oxygen reaction site,

the latter being in good agreement with prior work[12]. On the untransformed (110) surface, where the OER intermediates are coordinated by two La and one Ni-ion, the calculated overpotential is 0.79 V. A considerably lower overpotential of 0.46 V is obtained on the untransformed (111) Ni-facet, where the OER intermediates are coordinated by one La and one Ni-ion.

Since the transformed NiOO(H) surfaces are oxygen terminated (with each surface oxygen being coordinated by three Ni atoms), the OER can take place only via the oxygen vacancy site mechanism (OVSM). This mechanism belongs to the lattice oxygen mechanisms (LOM), but consists of the same reaction intermediates as the adsorbate evolution mechanism (AEM), starting at an oxygen vacancy in the surface layer (denoted by $^*$ in this work), with the $^*$O intermediate being in this case part of the lattice[39]. As described below in more detail, multiple inequivalent reaction sites emerge on the transformed surfaces. Fig. 6a shows the calculated overpotentials of the different sites for each untransformed and transformed facet. In the following, we discuss the overpotential of the most favorable reaction site for each facet. The transformed (001)-surface yields the highest overpotential of 0.79 V. For (110), partial coverage with hydroxyl groups at OER conditions leads to a considerable decrease of the overpotential from 0.58 V for a fully oxygen-terminated surface to 0.48 V for 1/3 ML H/NiOO/LaNiO$_3$(110). Since the (110)-facet shows the lowest tendency towards surface transformation, the actual overpotential is likely the overpotential of the untransformed perovskite (0.79 V). The overpotentials for the transformed (111) facet with two NiOO(H) layers are among the lowest for the different surfaces, with 0.45 V and 0.50 V at a LaO$_3$- and a Ni-terminated perovskite (111)-surface, respectively. Reducing the NiOOH thickness to a single layer yields higher overpotentials of 0.62 V and 0.66 V. The overall trend among the calculated overpotentials for the three orientations is consistent with the experimental findings. Figure 6b displays the OER reaction-free energies for the most beneficial sites for the transformed surfaces in each orientation, confirming the favorable binding energies and nearly equidistant $^*$OH → $^*$O and $^*$O → $^*$OOH steps, leading to the low overpotential of the (111) surfaces with two-layer oxyhydroxide-like coverage. At the transformed (110) surface, the OER intermediates show a similar step height. The higher overpotential for the transformed (001) compared to (111) stems from a weaker binding of the $^*$O intermediate. Consequently, $^*$OH → $^*$O is the

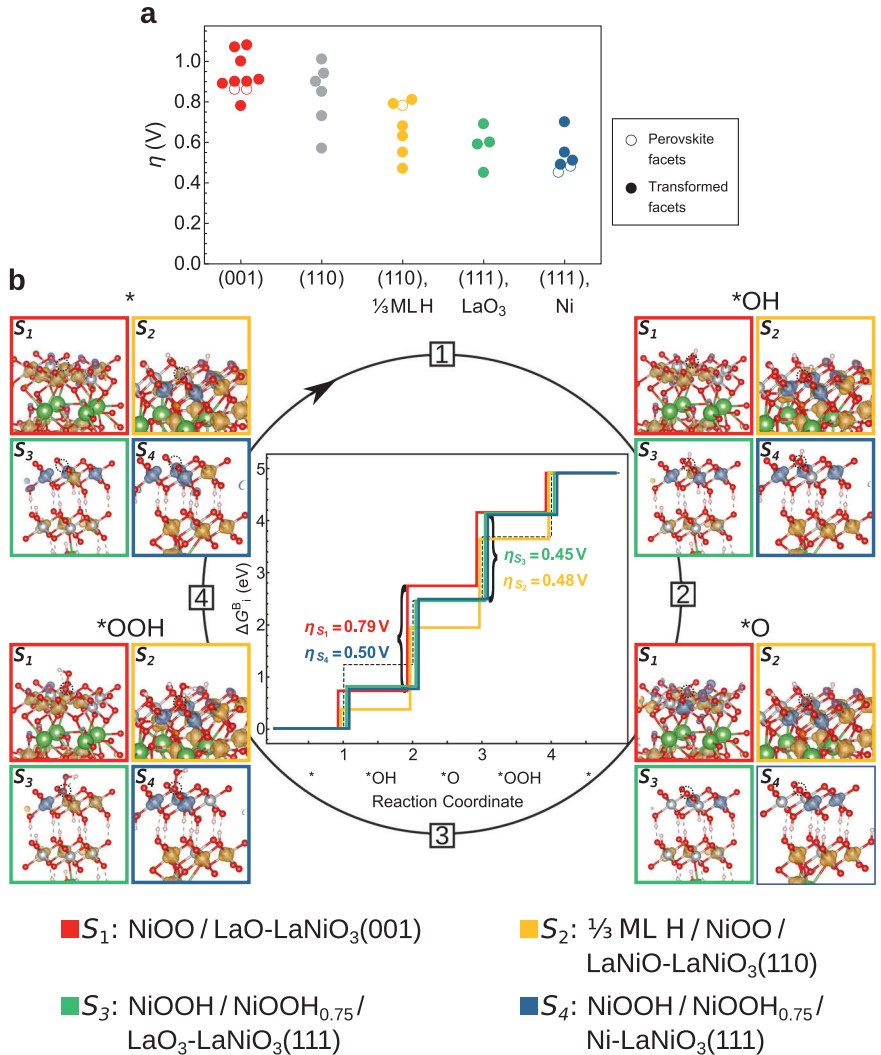

**Fig. 6 | Facet-dependent OER overpotentials of LaNiO3. a** Overpotentials for all inequivalent reaction sites on perovskite and transformed facets. For better visibility, points with similar $\eta$ are slighty shifted horizontally. **b** Reaction-free energies for the most favorable reaction site at the transformed surfaces. Side views of the intermediates, including spin density, are framed in the corresponding colors.

potential determining step (PDS) for the transformed (001)-facet, in contrast to a slight preference for $^*O \rightarrow {^*OOH}$ for the transformed (111) and (110) surfaces. The lower overpotential of the transformed (111) facet due to a later PDS indicates a lower Tafel slope, qualitatively consistent with the experimental trend (see Fig. 3f).

To shed light on the origin of distinct OER activity of the transformed (001), (110) and (111) facets, we now focus on their structural and electronic properties. As shown in Fig. 7, a striking feature is the substantial distortion of the NiOO layer on (001) and (110)-oriented LaNiO3 compared to (111), stemming from the lattice mismatch of the hexagonal NiOO overlayer and the square/rectangular perovskite facets. This leads to considerably increased Ni-O bond lengths between 2.0 Å and 2.5 Å along the [110] direction for (001) and along [100] for (110). In contrast, the NiOO(H) layer is much more regular at the (111) facet with a weaker elongation of few Ni-O bonds up to 2.0 Å due to a Jahn-Teller effect. Concerning the electronic and magnetic properties of the oxygen-terminated NiOO layer, a stripe arrangement of distinct Ni sites emerges: $Ni^{3+}$ with a magnetic moment of $\sim \pm 1\,\mu_B$ and $Ni^{4+}$ with a predominantly zero magnetic moment (a few $Ni^{4+}$ sites acquire a finite magnetic moment of $\sim 0.3-0.7\,\mu_B$ due to the lattice distortions). Overall, these variations in bond lengths and surrounding Ni sites lead to a high number of inequivalent reaction sites on the transformed surfaces, denoted in Fig. 7. This high irregularity

observed in the simulated supercells hints at a high degree of disorder in the real transformed catalyst. The presence of cations in high oxidation states has been recently proposed to activate surface oxygen[40]. Accordingly, several oxygen sites exhibit nonzero magnetic moment reaching up to $0.3\,\mu_B$, indicating the presence of holes.

While for $^*O$ we observe an equal ratio of $Ni^{3+}$ to $Ni^{4+}$ for all surface orientations (apart from the partially hydrogen-covered (110)), we encounter an overall increase of the magnetic $Ni^{3+}$ sites for the other intermediates during OER and also the presence of $Ni^{2+}$ at the (110) facet (with a magnetic moment of $\sim 1.5\,\mu_B$). The only exception emerges at the transformed (001) facet, where the number of $Ni^{3+}$ increases only by one for the vacant site ($^*$). For comparison, all Ni sites switch to 3+ for $^*$ at the transformed (111) surfaces. This difference explains the weak binding of the $^*O$ intermediate and the high overpotential of the transformed (001) facet. Despite the structural similarity between the (001) and (110) transformed facets, the partial hydroxylation of (110) and the connectivity to the LaNiO-terminated (110) facet lead to a substantially stronger binding of adsorbates than at the LaO-terminated LaNiO3(001). In summary, our DFT+$U$ investigation not only confirms the distinct OER activity of the three facets, but reveals significant structural differences resulting in a variation of the amount of magnetic Ni ions surrounding the active oxygen sites that impacts the binding energies of intermediates. This also establishes a link to the

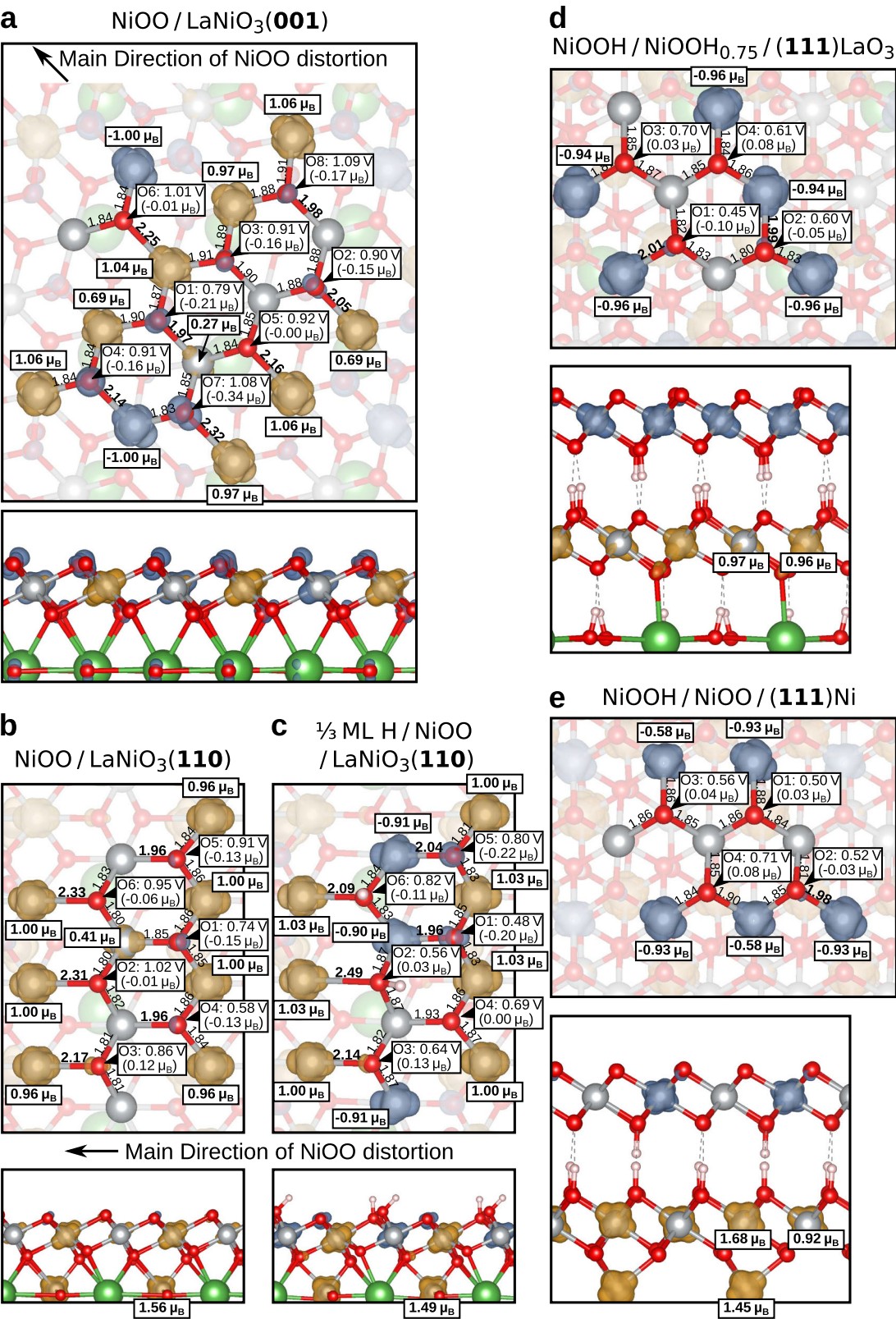

**Fig. 7 | Structures of the transformed facets and Ni oxidation states.** Top and side views of the transformed (**a**) (001), **b** (110), **c** partially covered (110) facets, as well as (**d**) (111) with an underlying LaO₃-termination and (**e**) (111) with an underlying Ni-termination. Blue and orange clouds represent minority and majority spin iso-surfaces. In the top views, to focus on the topmost oxygen and Ni layers, the underlying layers are shown semi-transparent. Ni-O lengths in Å are displayed, with substantially elongated bonds marked in bold. Ni magnetic moments in $\mu_B$ are shown in framed labels. For each oxygen reaction site, the overpotential and the magnetic moment are shown.

current discussion of the role of magnetism and orbital occupation in electrocatalysis[1,41].

## Discussion

In conclusion, we investigated the facet-dependent OER activity of epitaxially grown, Ni-terminated $LaNiO_{3-\delta}$ thin films. The OER overpotential of the (111) facet was found to be smaller than for the (110) and (001)-oriented surfaces. A redox peak at $\approx 1.4$ V, before the onset of OER, hints at a surface transformation into an oxyhydroxide-like NiOO(H) layer. This transformation is more pronounced on the (111) surfaces than for the (001) and (110) counterparts, involving at least two layers of Ni atoms for the (111) surface already during the first redox wave. X-ray absorption analysis shows that the transformation is indeed connected to a new surface phase with edge-sharing octahedra, which becomes more prominent after extended operation at extreme conditions. While we previously suggested a similar oxyhydroxide-like NiOO(H) as the active surface phase for the (001) facet[12], the DFT+$U$ simulations presented here for the three facets are not only consistent with the activity trend observed in the experiment, but reveal decisive differences between the transformed facets: In contrast to (111), a pronounced distortion of the NiOO(H) layer emerges on the transformed (001) and (110) surfaces due to a substantial lattice mismatch to the underlying perovskite. Despite the structural similarity between the latter transformed facets, the partial hydroxylation of (110) and the connectivity to the LaNiO-terminated (110) facet lead to a substantially stronger binding of adsorbates as well as a lower ovepotential. The limited transformation of the (110) facet observed in the experiment implies rather the presence of an untransformed and comparably inactive (110) perovskite surface. A detailed analysis reveals an enhanced fraction of magnetic $Ni^{3+}$ as the origin of the improved activity of the (111) facet.

Our combined experimental and computational investigation provides new insight into the nature of the transformed surface and surface transformation pathways, namely how they depend on the crystallographic facets, yielding an explanation of the facet-dependent activity trends of current interest. The facet-dependent electronic and atomic structure of the transformed surfaces, in turn, dictates the binding of reaction intermediates, and therefore the electrocatalytic activity. Our study thus highlights that the crystal facet of the original surface affects the physical and catalytic properties of the transformed surface, which offers an avenue to optimize the active surface phase under reaction conditions. Therefore, detailed understanding and control of faceting and facet transformations during the reaction is essential for the rational design of electrocatalysts for the energy and green chemistry sector.

## Methods

### Epitaxy of $LaNiO_{3-\delta}$ thin films

Epitaxial 20 nm thick $LaNiO_{3-\delta}$ thin films were deposited on epipolished $SrTiO_3$ single-crystal substrates with (001), (110) and (111) orientation (Shinkosha Co., Ltd., Japan) by reflection high-energy electron diffraction-controlled pulsed laser deposition (SURFACE systems+technology GmbH & Co. KG, Germany). Here, a KrF-excimer laser (Lambda Physik Lasertechnik, Germany) with a wavelength of $\lambda = 248$ nm was operated, using a repetition rate of $f = 5$ Hz and a laser fluence of $F = 1.60$ J/$cm^2$. The growth temperature was $T = 550$ °C and the oxygen partial pressure was $p(O_2) = 0.02$ mbar. A target-to-substrate distance of 60 mm was used. After deposition, the samples were annealed at $p(O_2) = 0.1$ mbar for 10 min and finally quenched.

### Physical and electrochemical characterization

The surface morphology was imaged using a Cypher SPM atomic force microscope (Research Asylum, Germany), operated in tapping mode. Crystallographic properties were investigated with a D8 ADVANCE diffractometer (Bruker AXS GmbH, Germany) with a monochromized Cu K$_\alpha$ X-ray source.

X-ray photoelectron spectroscopy measurements were performed using a Omicron XM 1000 Al K$_\alpha$ monochromated X-ray source (1486.6 eV, FWHM = 0.26 eV) and an Omicron EA 125 energy analyzer and with a Phi 5000 VersaProbe, ULVAC Phi, Physical Electronics Inc. The mean escape depth $d$ is defined through the inelastic mean free path of photoelectrons $\lambda = 2.2$ nm (calculated via QUASES-IMFP-TPP2M)[42] and the photoemission angle $\theta$ through $d = \lambda \times \cos \theta$[43]. The photoemission angles were 0° and 55°. To estimate the surface stoichiometry from the measured intensities, we compared the relative intensity of the A-site and B-site peaks as a function of $d$. During measurements the pressure was smaller than $3 \times 10^{-10}$ mbar.

The setup for characterizing the catalytic activity and lifetime consisted of a potentiostat (Bio-Logic Science Instruments, France) and a rotating disc electrode (Pine Research, USA) with a custom-made adapter for the electrocatalysts. We followed the recommended best practices in[32]. For sufficient electronic contact between the sample and Pt area inside the adapter, at the edges of the thin films and on the backside 50 nm thick Pt contacts were sputtered. As electrolyte, 99.99% KOH pellets (Sigma-Aldrich Chemie GmbH, Germany) were dissolved in de-ionized water to get a 0.1 M KOH solution. Next to the thin film electrode, a Hg/HgO electrode (CH Instruments, USA), calibrated to a reversible hydrogen electrode (HydroFlex, USA) aforehand, was used as reference electrode and a Pt wire as counter electrode and all were immersed in the $O_2$ saturated electrolyte. A sample surface area with diameter of 0.75 cm, limited by an FFKM O-ring (ERIKS Deutschland GmbH, Germany), was exposed to the solution.

Electrochemical experiments reported in the main text were done in a glovebox under a nitrogen and oxygen atmosphere and performed at room temperature. A rotation rate of 1600 rpm was applied to the disc electrode. Since the catalysts showed an uncompensated series resistance $R_U$, electrochemical impedance spectroscopy was applied for $iR$-correction. $R_U$ was extracted from the intercept of the high-frequency impedance data with the Re(Z) axis using linear extrapolation. Differences in uncompensated resistance are related to small changes in placement of the reference electrode in the cell, rather than resistivity differences across different orientations. We note that for reporting absolute values of the OER activity, the $iR_S$ correction rate and procedure may have a large effect[44]. But we emphasize that our conclusions rely on one-to-one comparison of model systems, where the same activity trends are observed with and without $iR$-correction. For testing the catalytic lifetime, chronopotentiometry was performed. Here, a quasi-static staircase was used to ramp up the current density up to 3.2 mA/$cm^2$ and to determine the Tafel plot under steady-state conditions (each current density step was kept for 10 min). Quantification of charge during cyclic voltammetry was performed following refs. 31,45. The amount of charge was calculated using the EC lab software package by integration of the anodic peak after subtraction of a linear background (capacitive contribution).

### X-ray absorption spectroscopy

X-ray absorption spectroscopy (XAS) spectra at Ni-K edge were collected at the KMC-3 beamline[46] at the synchroton BESSY II (Helmholtz-Zentrum Berlin für Materialien und Energie). Spectra were recorded in fluorescence mode using a 13-element silicon drift detector (SDD) from RaySpec. The used monochromator was a double-crystal Si (111), and the polarization of the beam was horizontal. NiO (ROTI®METIC 99.999%) and $LiNiO_2$ (Sigma-Aldrich ≥98%) reference samples were prepared by dispersing a thin and homogeneous layer of the ground powder on Kapton tape. After removing the excess material, the tape was sealed, and the excess of Kapton was folded several times to get 1 cm × 1 cm windows. The reference materials $LiNiO_2$ and NiO were measured during the same beamtime with identical parameters of the monochromator where the energy scale of $LiNiO_2$ was calibrated using NiO as standard. The NiO standard and the $LaNiO_{3-\delta}$ samples were energy calibrated using a Ni metal foil (GoodFellow 99.9%) (fitted reference energy of 8333

eV in the first derivative spectrum) with an accuracy ± 0.1 eV. At least three scans of each sample were collected to $k = 14$ Å$^{-1}$. All spectra were normalized as described in detail in Villalobos et al.[47] In short, a straight line obtained by fitting the data before the K edge was firstly subtracted, followed by a division by a polynomial function obtained by fitting the data after the K edge. The FT of the EXAFS was calculated between 35 and 550 eV above the Ni-K edge ($E_0 = 8333$ eV) with a cosine window covering the first and last 10% of the data range. EXAFS simulations were performed using the software SimXLite. Phase functions were calculated with the FEFF8-Lite program (version 8.5.3, self-consistent field option activated)[48]. Atomic coordinates of the FEFF input files were generated from the (111)-oriented LaNiO$_{3-\delta}$ (Fig. 4; identical to those used for DFT) and NiO structures[49]. The data range used in the simulation was 35 - 550 eV (3.03 - 12.01 Å$^{-1}$) above the Ni-K edge ($E_0 = 8333$ eV). For the pristine and 1 h CA samples, two Ni-La with coordination numbers of 2 and 6 were used[36]. Whereas for the 16 h CA sample, a combination of two sets of phase functions was used (a set for the LaNiO$_{3-\delta}$ phase and a set for the NiO phase). For all samples, an extra Ni-O shell was added at higher distances (around 4.1 Å), which significantly improved the fits (error sum and goodness-of-fit indicator R-factor). Different alternatives were tested for the EXAFS fit, including Ni-O shell from the perovskite structure, one Ni-La shell instead of two and fits without the second Ni-O shell. The best fits with the lowest R-factors were obtained with the parameters described above. An amplitude reduction factor ($S_0^2$) of 0.7 was used[50]. The Debye-Waller parameters for all shells were set to 0.05 Å to avoid overparameterization. The EXAFS simulations were optimized by the minimization of the error sum obtained by summation of the squared deviations between measured and simulated values (least-squares fit). The parameter errors were obtained as described in detail in Ref. 51 using Fourier-filtered data between 0 and 4.5 Å$^{-1}$.

### Density functional theory calculations

Density functional theory calculations were carried out using the projector augmented wave method and pseudopotentials[52,53], as implemented the Vienna ab-initio simulation package (VASP)[54–56]. For the exchange-correlation functional the generalized gradient approximation in the implementation of Perdew, Burke and Ernzerhof (PBE)[57] was used. Static correlations were taken into account by applying an effective Hubbard-parameter $U_{\mathrm{eff}} = U - J$ on Ni 3$d$-states within the Dudarev approach[58]. For the perovskite surfaces $U_{\mathrm{eff}} = 2.0$ eV was used, which was determined previously by comparing the band structure to photoemission spectroscopy[12]. For the transformed, oxyhydroxide-like surfaces a higher value $U_{\mathrm{eff}} = 5.5$ eV is necessary, which was previously calculated via linear response theory for bulk NiOOH oxyhydroxide[59]. An in-depth discussion of the effect of the Hubbard $U$ parameter on the OER reaction intermediate binding energies and overpotential is provided in the Supplementary Note 5. For convenience, $U_{\mathrm{eff}}$ is denoted as $U$ in the main text and Supplementary Information. The plane-wave basis set cut off is 520 eV.

Pristine and transformed LaNiO$_3$(001) surfaces were modeled by a NiO$_2$-terminated $2 \times 2 \times 8$ slab and a (LaO)-terminated $2 \times 2 \times 7$ slab with an additional oxyhydroxide-like layer (containing 8 Ni ions) on top, respectively. This resulted in a total slab thickness ≥15.7 Å. Pristine and transformed LaNiO$_3$(110) surfaces were modeled by a LaNiO-terminated $2 \times 2 \times 9$ slab (the other possible (110) termination, O$_2$, was treated as a surface coverage) and a LaNiO-terminated $2 \times 2 \times 9$ slab with an additional oxyhydroxide-like layer (containing 12 Ni atoms) on top, respectively. This resulted in a total slab thickness ≥14.7 Å. The LaNiO$_3$(111) surfaces were modeled by a Ni-terminated $2 \times 2 \times 14$ slab for the perovskite surface and LaO$_3$-terminated $1 \times 1 \times 13$ and Ni-terminated $1 \times 1 \times 12$ slabs with additional oxyhydroxide-like layer(s) (containing 4 Ni ions). This resulted in a total slab thickness up to 22.2 Å. A vacuum layer of at least 11.8 Å was used to separate the slabs from their periodic images and a dipole-correction in the vertical direction was applied due to the asymmetric setup. For all systems, the lateral lattice constant was

set to the one of SrTiO$_3$ to model films grown on this substrate. For (001)- and (110)-oriented systems, a $4 \times 4 \times 1$ and $3 \times 4 \times 1$ Monkhorst-Pack[60] $k$-mesh was used, respectively, and for (111)-oriented systems, a $4 \times 4 \times 1$ Γ-centered $k$-point mesh. The ionic positions in the bottom four layers of the (001)-oriented slabs, the bottom three layers of the (110)-oriented slabs and the bottom six layers of the (111)-oriented slabs were fixed to the bulk values. Structural relaxation was performed until the residual forces were below 0.02 eV/Å. Different initial magnetic configurations (ferro- and antiferromagnetic and quenched magnetic moments) were explored which converged to the same final configuration.

The oxygen evolution reaction was modeled in the computational hydrogen electrode approach, proposed by Nørskov and Rossmeisl[61,62], where the OER is split into four consecutive reaction steps, the reaction free energies result from differences in binding energies $\Delta G^B$. The binding energies of the intermediates, as well as coverages with different functional groups for the construction of a surface Pourbaix diagram can be described by the following generalized formula:

$$
\begin{aligned}
\Delta G^B = E_* - E_{\mathrm{ads}} &+ N_{\cdot \mathrm{OH}} \cdot \left( \frac{1}{2} H_{\mathrm{H}_2} - H_{\mathrm{H}_2\mathrm{O}} + \Delta(\Delta G - \Delta E)_{\cdot \mathrm{OH}} \right) \\
&+ N_{\cdot \mathrm{O}} \cdot \left( H_{\mathrm{H}_2} - H_{\mathrm{H}_2\mathrm{O}} + \Delta(\Delta G - \Delta E)_{\cdot \mathrm{O}} \right) \\
&+ N_{\cdot \mathrm{OOH}} \cdot \left( \frac{3}{2} H_{\mathrm{H}_2} - 3 H_{\mathrm{H}_2\mathrm{O}} + \Delta(\Delta G - \Delta E)_{\cdot \mathrm{OOH}} \right) \\
&+ (1 \cdot N_{\cdot \mathrm{OH}} + 2 \cdot N_{\cdot \mathrm{O}} + 3 \cdot N_{\cdot \mathrm{OOH}}) \cdot (e \cdot \phi - \mathrm{k}_B T \cdot 2.3 \cdot \mathrm{pH})
\end{aligned}
\tag{1}
$$

where $E_*$ and $E_{\mathrm{ads}}$ are the DFT total energies of the slab without and with adsorbates, $N_i$ the respective number of functional groups, $H_{\mathrm{H}_2}$ and $H_{\mathrm{H}_2\mathrm{O}}$ the formation enthalpies of the respective molecules, and $\phi$ is the applied potential. The vibrational correction term $\Delta(\Delta G - \Delta E)$ consists of the change in zero-point energy ZPE, specific heat contribution $C_V \cdot T$ and entropic contribution $T \cdot \Delta S$ (both at room temperature $T = 298$ K in this work) upon adsorption. For the OER on NiOO/LaNiO$_3$(001), which takes place via the OVSM lattice oxygen mechanism, as decribed in the main text, these quantities were obtained from phonon frequencies calculated (via finite differences) with the post-processing tool vaspkit[63], relative to molecular H$_2$O at $T = 298$ K and $p = 0.035$ bar (where gaseous and liquid phases are in equilibrium) and H$_2$ at standard conditions ($T = 298$ K and $p = 1$ bar). The obtained values, listed in Supplementary Tables 14 and 15, are in good agreement with Ref. 12. Since there are no results available for hydrogen intercalated between NiOO and the underlying perovskite, we have explicitly calculated the vibrational corrections for O and OH located between the NiO(OH$_x$) layer and LaNiO$_3$(111) (see Supplementary Table 16). For intercalated H, the correction term is 95 meV lower in energy compared to H adsorbed on the surface.

Occupancies of Ni 3$d$ orbitals, from which the formal oxidation states were determined according to the procedure outlined in Sit et al.[64].

## Data availability

The experimental and computational data that support the findings of this study are available in the 4TU.ResearchData repository at https://doi.org/10.4121/2da7626a-00a6-4f4f-95b1-d9ccbfbe7993 and in the NOMAD repository at https://doi.org/10.17172/NOMAD/2023.11.17-1[65,66].

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

## Acknowledgements

R.P. and A.F. acknowledge funding by the German Research Foundation within CRC/TRR247 (project number 388390466, B04) and computational time at the Gauss Center, Leibniz Rechenzentrum within project pr87ro. E.K. acknowledges funding by the Netherlands Ministry of Economic Affairs' Top Consortia for Knowledge and Innovation (TKIs) Allowance. C.B. has received co-funding from the European Union (ERC, 101040669 - Interfaces at Work). Views and opinions expressed are however those of the author(s) only and do not necessarily reflect those of the European Union or the European Research Council. Neither the European Union nor the granting authority can be held responsible for them. F.G. acknowledges funding by the German Research Foundation in the framework of the SPP 2080, project no 493705276 (GU1604/4). M.R. and D.A. have received funding for this project from the European Research Council (ERC) under the European Union's Horizon 2020 research and innovation program under grant agreement No. 804092. We thank the Helmholtz-Zentrum Berlin für Materialien und Energie for the allocation of synchrotron radiation beamtime. The XAS experiments were financially supported by funds allocated to Prof. Holger Dau (Freie Univ. Berlin) by the Bundesministerium für Bildung und Forschung (BMBF, 05K19KE1, OPERANDO-XAS) and by the Deutsche Forschungsgemeinschaft (DFG, German Research Foundation) under Germany's Excellence Strategy–EXC 2008–390540038–UniSysCat. The authors thank Melissa Goodwin for FIB sample preparation and Lidewij M.A. Krakers, Martina Tsvetanova, Dulce M. Morales and Joaquín Morales-Santilices for discussion and help with data collection as well as Michael Haumann, Paul Beyer and Götz Schuck for support at the beamline.

## Author contributions

A.F. and R.P. conceived, designed and interpreted the DFT simulations, performed and analyzed by A.F.; F.G. and C.B. contributed through in-depth discussion and comparison to the experiments during all stages. F.G. and C.B. conceived and designed the experiments. M.A.W., E.M. and E.M.K. prepared the samples and performed thin-film and electrochemical characterization. D.A., E.M., J.V. and M.R. performed the X-ray absorption experiments. D.A. and M.R. modelled, analyzed and interpreted the spectroscopy results; A.F., R.P. and C.B. wrote the manuscript with contributions from all authors.

## Funding

## Competing interests

The authors declare no competing interests.
