## [Peer Review File · Nature Communications]

REVIEWER COMMENTS

Reviewer #1 (Remarks to the Author):

The authors investigate the (001), (110) and (111) facet of $\text{LaNiO}_{3-\delta}$ electrocatalysts and reveal a facet dependent OER activity with spectroscopic characterizations and DFT+U calculations: the (111) facet has the highest observed activity and stability, and the most pronounced Ni redox features. Overall, this experiment was well designed, and the paper was well written. The results are interesting and will be helpful for future design of electrocatalysts. As the authors stated that similar results have been claimed in their previous publications [12, 22], what is the innovative finding in this work? Also, some concerns below should be addressed for re-consideration.

1. OER activity over different facets was apparently compared from the CV curves (Figure 3). For a fundamental study, it would be more meaningful to get normalized with the number of active site, namely Ni on different facets in this study, to exclude the improvement caused by the increase of Ni spots.
2. More characterizations after 16hr should be performed, like surface microscopy and XRD to further confirm the changes in morphology, structure, and even stability coming from surface reconstruction.
3. In calculation, the authors modeled OER in terms of thickness and proton intercalation. It will be more experimentally convincing with a depth profile study like XPS to give clearer concept about how stable the sample was, how thick the sample layer was forced to restructure during OER, and how possible the proton intercalated and interacted at the interface.
4. In Figure 4b, I agree that the peak iii disappeared in sample 16h-CA. But I do not think the peak ii is new because it stays over there exactly same as the as-prepared and 1h CA samples. I hope an expert in XAS can give more comment on this issue.
5. In page 5, all Figure 3bs should be Figure 3a.

Reviewer #2 (Remarks to the Author):

In this work, F' ungerlings et al., studied the (001), (110) and (111) facet of $\text{LaNiO}_{3-\delta}$ electrocatalysts and revealed a facet- dependent activity by using combined electrochemical characterization and

theoretical simulation approaches. They found that both (001) and (111) facets undergo a significant surface transformation into oxyhydroxide-like NiOO and the (111) facets obtain deeper transformation, which delivers best OER performance. The author employed systematic DFT+U simulation to confirm that the lattice parameter of transformed layer matches better with the pristine (111) slabs, leading to the less lattice distortion and optimized binding energy of reaction intermediates. Actually, previously, the work regarding facet dependent OER activity of LaNiO₃ has been reported by same group (Frontier Chem, 2022, 10, 913419), and the major novelty of this work focuses on the facet dependent on-site phase transformation behavior including their dynamic protonation and lattice mismatch. I will not further consider the publication of this work until the author addresses following questions.

(1) First of all, the author claimed that (111) facet exhibits best activity and stability. However, the discussion about the relationship between facet dependent phase transformation and durability is poor. To be specific, I am curious about the worst stability and activity of (110) facet, as compared to (111) and (100) counterparts. Since (110) facet has been regarded as the one having least phase transformation, it intuitively indicates that this facet is more stable or electrocatalytic inert. Why does it decay so fast? What is the mechanism behind the difference in stability? The authors are required to clarify it by offering more convincing data and explanation.

(2) Since (001) and (111) could be terminated by NiO₂/LaO and Ni³⁺ /LaO₃, respectively. Do the author mean most of the (001) and (111) facets were terminated by NiO₂ and Ni³⁺, respectively? I am glad to see that the authors provide surface angle-resolved XPS to confirm the surface rich phase of Ni species. If so, why does the author still persist on constructing the models with LaO and LaO₃ termination which is literally impossible for (001) and (111) facets?

(3) How does the author calculate, quantify or expect the amount the charge layer based on CV scan shown in Figure 3a? the detail calculation method should be given. Also, why does the author choose the fast scan rate of 500 mv/s ?

(4) There are some typos all through the manuscript, suggesting the carelessness of proofreading. For example, there is no inset in figure 3b. it should be in figure 3a. Only part of the EIS plot can be found in figure 3b. Detailed analysis and fitting of is lacking.

(5) In Figure S4 of Pourbaix plot, the authors claim that at the untransformed, Ni-terminated surface of LaNiO₃(111), two coverages are close in energy under reaction conditions pH = 13 and $\varphi = 1.68$ V versus RHE, either one monolayer (ML) *OOH or 0.75 ML *OH+0.25 ML *OOH. However, as long as we go back to Figure S4a, we can easily find that there is no unclear margin at this condition. Instead, the one monolayer OOH is predominant. I thought the author wished to talk about Figure S4b in which both scenarios are possible.

(6) There have no TEM images to directly visualize the phase transformation, which are essentially required to confirm the reconstruction depth. In XANES plot, the spectra of samples after 16-hour anodic treatment are provided. Why is 16 h? Is 16 h long enough to realize the complete reconstruction? What if further extend the treatment time? To answer this set of questions, I would love to know the performance of the electrocatalyst after 16 h treatment first.

(7) Usually, hard X-Ray XAS is a bulk sensitive characterization tool. Based on the EXAFS data, I can expect the deep reconstruction of (111) facet. However, the spectra for (100) and (110) are missing, which are required to show for fair comparison. Moreover, a table of fitting parameter should be given.

(8) Basically, the theoretical simulation part of this work is well-organized. I just have one more question regarding the calculation of reaction Gibbs free energy. As the author claims: "At the transformed oxygen-terminated LaNiO₃ surfaces, the OER cannot take place on top of cation sites like on the perovskite surfaces, but is performed via the lattice oxygen mechanism at a surface oxygen site." Usually, the absorbent evolution mechanism happens on metal site only with the reaction intermediate of OH, O and OOH. However, for lattice oxygen mechanism, the reaction coordinate is different. Nevertheless, it seems like the intermediate for AEM is directly used for LOM, as shown in Figure 6. Please explain it.

Reviewer #3 (Remarks to the Author):

The manuscript reported by Pentcheva et al. investigated the crystal-facet-dependent surface transformation and contribution to the OER activity of LaNiO_{3-δ} from both the perspective of the experiment and DFT+U calculation. The results demonstrated that the (111) facet possesses a lower overpotential than both (001) and (111) facets. The reasons can be ascribed to that the transformed surface is thicker for (111) than for (001) and exhibits a protonated interface, as well as a better match between the transformed surface and the underlying perovskite. This leads to a favorable binding energy of intermediates. Overall, this work is well-designed and novelty. But a major revision is recommended before considering publication in Nature Comm. Detailed suggestions are as follows:

(1) The structure and content of the manuscript need to be revised and simplified, especially in DFT section. Only the most relevant results should be present in the maintext, because the current analysis and expression are ambiguous.

(2) (001), (110), and (111) facets are experimented, why only both (001) and (111) facets are considered in DFT calculation? The DFT calculation based on (110) facet needs to be supplemented. Because these results are essential to confirm the consistent conclusion with the experiment that (111) is favorable to (001) and (110) facets.

(3) In Fig. 3b, why about 20 Ω uncompensated resistance difference between (001) and (111) facets are tested in Nyquist impedance plots? The facets maybe not dictate such huge solution resistance differences.

(4) The applied iR-correction rate should be highlighted for its huge effect on OER results. The relevant reports (Energy Environ. Sci. 11 (2018) 744-771; Materials Today Energy 32 (2023) 101246) have highlighted this point.

(5) A thicker oxyhydroxide-like NiOO surface transformation in (111) facet maybe increase the ECSA. Thus, ECSA test can consider added.

(6) A further long-term galvanostatic hold (such as ≥ 100 h) would induce a full of surface oxyhydroxide transformation, at that moment, will the similar conclusion that (111) is favorable to (001) and (110) be obtained?

Reviewer 1 (Remarks to the Author):

Comment: The authors investigate the (001), (110) and (111) facet of $\text{LaNiO}_{3-\delta}$ electrocatalysts and reveal a facet dependent OER activity with spectroscopic characterizations and DFT+U calculations: the (111) facet has the highest observed activity and stability, and the most pronounced Ni redox features. Overall, this experiment was well designed, and the paper was well written. The results are interesting and will be helpful for future design of electrocatalysts. As the authors stated that similar results have been claimed in their previous publications [12, 22], what is the innovative finding in this work? Also, some concerns below should be addressed for re-consideration.

Response: We thank the referee for the overall very positive assessment and recommendation of our manuscript.

Novelty: Surface transformations such as we discovered for the single LaNiO_3 (100) facet in Ref. [12] represent a key aspect defining catalyst functionality, as exemplified by the recent work from the Zhang group (<https://doi.org/10.1039/D2SC07034K>). Nonetheless, such chemical transformations are vastly different across different materials and a fundamental understanding of these phenomena is currently lacking. **Here we demonstrate and explain that surface transformations even drastically vary from facet to facet within a given material.** As acknowledged by reviewer 2, the significant novelty of our study thus lies in **the deep mechanistic insights into the relation of surface phase formation in different crystal orientations**, underlying electronic properties and OER activity from the comprehensive DFT+U calculations, combined with further detailed experimental characterization, as opposed to reporting a particularly active or inactive facet.

Ref. [22] is merely a perspective paper on general relationships of activity and stability in OER catalysts, in which we mention activity-stability data for different facets of perovskite oxides, LaNiO_3 being only one example, in a phenomenological compilation of electrochemical data to illustrate a general trend. Ref. [22] does not at all address any detailed discussion on electrochemical behavior of LaNiO_3 in different crystal orientations, nor elaborates on the underlying mechanistic processes that lead to the observed activity and stability. In fact, Ref. [22] directly emphasizes the (at that point) lack of mechanistic understanding, which we now resolve based on an advanced experimental and theoretical investigation. While Ref. [22] did not provide (nor attempt) a mechanistic understanding, our present manuscript fills the research gap we identified in our perspective paper and provides a significant scientific advance in understanding structure-activity relationships in OER catalysts.

We have added/adapted the following statements to highlight the novelty of our manuscript. In addition, we have edited to introductory part to further improve the flow of this section.

Changes to the text (changes highlighted in yellow):

“We find that the underlying perovskite facet affects greatly the structure and bonding, i.e. a significant distortion occurs at (001) and (110) due to lattice mismatch. This influences the ratio of Ni^{2+} and Ni^{3+} to Ni^{4+} and by this the binding energy of reaction intermediates, resulting in the three distinct OER activities of the transformed facets”

...

“Phenomenologically, different crystal facets were reported to yield different activity and stability. However, any deeper understanding of the role of facet-dependent surface-transformations of highly dynamic surfaces like $\text{LaNiO}_{3-\delta}$ and the resulting surface-structure under reaction conditions remains elusive.”

...

“Interestingly, we also found the lifetime of the $\text{LaNiO}_{3-\delta}$ (111) facet increased compared to the (110) and (001) orientations, as we reported in reference [22], where we phenomenologically compared the lifetimes of differently oriented $\text{LaNiO}_{3-\delta}$ thin films via chronopotentiometry.”

Comment: 1. OER activity over different facets was apparently compared from the CV curves (Figure 3). For a fundamental study, it would be more meaningful to get normalized with the number of active site, namely Ni on different facets in this study, to exclude the improvement caused by the increase of Ni spots.

Response: The referee raises an interesting point, requesting to normalize the activity with respect to the number of Ni ions in each orientation. We are happy to provide the additional analysis as suggested. We now plot the turnover frequency (TOF) obtained by normalization of the current density to the number of Ni ions in a perfect perovskite surface, see Response Figure 1. In this normalization, the activity differences appear even larger, because the number of Ni ions per unit area increases from (111) facet to (110) to (100), with 3.79, 4.64, and 6.56 Ni ions per nm^2 . However, we note that the normalization to the number of ions in the perfect surface is in our opinion not suitable to assess the intrinsic activity because of the observed phase transformation. The details in the atomic arrangement lead to an inherent error in any estimate of the experimental activity normalized to the number of “active sites”, even if we were to attempt normalizing to number of Ni ions in the suspected transformed surface. However, we note that the activity trend also holds when normalizing to the peak area of the Ni oxidation/reduction peaks. We included the TOF data in the SI and mention the observation in the main text.

Response Figure 1: Left: Cyclic voltammetry in the OER potential regime, showing the average between anodic and cathodic sweep of the second cycle (Fig. 3c in the main text). Right: turnover frequency (TOF) obtained by normalization of the current density to the number of Ni ions in a perfect perovskite surface.

Addition to the text: “The activity trend is even more pronounced when comparing the turnover frequency (TOF) obtained by normalization of the current density to the number of Ni ions in a perfect perovskite surface (Supplementary Fig. 4).”

Comment: 2. More characterizations after 16hr should be performed, like surface microscopy and XRD to further confirm the changes in morphology, structure, and even stability coming from surface reconstruction.

Response: We appreciate the referee’s suggestion and performed additional experiments for up to 100 h of operation. However, we point out that the key message of our manuscript is not the long-term degradation behavior but the transformation occurring in the initial operation of LaNiO₃ electrocatalysts as a function of surface orientation.

The main implication of the additional experiments is: **The stability of the epitaxial thin films is fully sufficient to draw the conclusions presented in the manuscript.**

In order to address the reviewer request, additional data is provided in the revised manuscript. For clarity, we structure these new data into electrochemical data, XPS, TEM, XRD and XRR after different operation times. The data is included as Supplementary Figs. 7-9 of the revised manuscript.

Response Figure 2: Chronoamperometry (CA) at 1.8 V vs. RHE of (111) LaNiO₃ for 100 h.

Electrochemical data: During the 100 h operation time (to our knowledge one of the longest activity tests reported for epitaxial thin film model systems), the activity of the (111) LaNiO₃ remains high: The film undergoes an initial transformation (with a small dip in activity followed by an activity enhancement by factor ~2). After a few hours, the activity remains comparably constant over the measurement time.

XPS: As we stated in the caption of SI Fig 3 of the previous submission, “Only minor changes in cation composition of the surface layer is observed.” We now add additional information on the surface vs. subsurface composition, obtained from angle dependent XPS measurements (resulting in different mean information depth $d=1.1$ nm and $d=2.1$ nm). The overall Ni/La ratio at $d=1.1$ nm is higher than for $d=2.1$ nm for all samples, indicating predominant Ni termination in the as-prepared state for both (100) and (111) orientations. For operation up to 16h, the Ni/La ratio remains constant, as shown exemplarily for the (111) orientation in Response Figure 3. This indicates that on average, the transformed surface is as Ni-rich as the predominantly Ni-terminated as-prepared state.

Only after additional stability testing up to a total operation time of 100 h, we find that for both (100) and (111) samples, the Ni/La ratio changes: While the surface remains slightly Ni-rich, the overall

cation ratio approaches 1:1. But we note that the XPS signal intensity decreased by a factor of roughly 10 from the pristine state to the 100h CA state, indicative of an increased surface roughening (in line with the XRR and TEM results discussed below), making Ni/La ratios extracted from the area-integrated technique less reliable.

Response Figure 3: a) Cation ratio from XPS for a (111) LaNiO₃ sample as a function of CA time (CA interrupted for intermediate XPS measurements). b) Cation ratio from XPS for a (001) LaNiO₃ sample, with a continuous CA measurement for 100 h. Different probing depths d are obtained with different photoemission angles (see manuscript).

TEM: We performed HAADF STEM analysis of (001) and (111) LaNiO₃ samples before and after 100 h CA, to track the most severe changes during stability tests (Response Figure 4). We note that cross-sectional TEM relies on preparation of lamella using a focussed ion beam (FIB). While we used a protective carbon layer during lamella preparation, the top surface (~1-2 unit cells) may be damaged during FIB, implying that TEM is not suitable to test the hypothesis of oxyhydroxide-like surface layers, which was the motivation for our focus on XAS in the original submission. Nevertheless, the TEM analysis provides valuable additional information: Even after 100 h of operation, the bulk of the film remained intact (in line with the XRD analysis below). Near the surface, we observe pit-like corrosion: The entire lamellas for both orientations were decorated with nm-sized amorphous regions with a depth of up to 3 nm. Within experimental resolution, the EDS composition of these regions and the surface is identical to the bulk of the film. These observations have important implications for stability-investigations of perovskite-type electrocatalysts and will form the basis for future studies focusing on the stability and deactivation mechanism rather than the initial transformation investigated in our manuscript.

Response Figure 4: HAADF STEM analysis of LaNiO₃ samples after 100 h CA for (001) orientation (a, c) and (111) orientation (b,d). Pit-like corrosion is observed after this long-time stability tests (highlighted with red and yellow ellipses in panels a and b). These pits do not result from the FIB preparation; they are absent for lamella prepared from as-prepared (001) and (111) (new Supplementary Fig. 2). LaNiO₃. In addition, we observe extended defects in the bulk of the film, for samples both before and after CA (examples highlighted with red arrows). These were observed before (ref. 12 of the main text), connected to the chosen synthesis temperature, which was in turn necessary to obtain the desired Ni-termination in the as-prepared state.

XRD/XRR: We performed X-ray reflectivity (XRR) and X-ray diffraction (XRD) before and after 100 h CA (See Response Figure 5). Overall, the data confirm that the bulk of the films did not change to a large extent, in line with the TEM analysis. The XRR data is consistent with an increased surface roughening and a similar thickness (within 1 μc) for all films. The nm-sized amorphous regions observed in TEM make the extraction of quantitative figures from XRR difficult. The (bulk-sensitive) XRD data indicates a similarly high crystallinity (similarly pronounced Laue fringes) and similar lattice constant before and after the stability test, with small thickness variations (variations in the period of the Laue fringes).

Response Figure 5: XRR and XRD analysis before and after 100 h CA for (001) orientation (a, c) and (111) orientation (b,d).

Conclusion: We observe local near-surface amorphization after extended stability tests, but otherwise the LaNiO_3 samples remain highly-ordered model systems in the bulk. We conclude that the long-time stability observed for the bulk of the films (XRD, TEM) and the comparably stable surface cation ratio (for at least 16 h of operation) support the hypotheses in our manuscript. We added this information to the new Supplementary Note 3. All of these observations are in line with the similarity of the XANES and EXAFS signatures of the (111)-oriented films after 1h operation (Fig. 4 in the main text).

Comment: 3. In calculation, the authors modeled OER in terms of thickness and proton intercalation. It will be more experimentally convincing with a depth profile study like XPS to give clearer concept about how stable the sample was, how thick the sample layer was forced to restructure during OER, and how possible the proton intercalated and interacted at the interface.

Response: The additional XPS analysis is part of our response to comment 2. As discussed above, area-integrating techniques may not be the ideal tool to track the surface chemistry, but our additional experiments and analysis indicate that under reasonable applied potentials, only the surface layer transforms, while the bulk remains intact and chemically unaltered within experimental resolution.

Experimentally, our best available approach for characterizing the layer thicknesses was based on “counting” the number of Ni ions participating in the redox reaction via cyclic voltammetry, shown as the inset of Figure 3a, and discussed in the main text. This is a well-established method to measure

the number of electrochemically accessible metal cations (10.1021/accounts.mr.1c00087). In the revised manuscript, we highlight this aspect in greater detail.

Changes to the text (changes highlighted in yellow): “The area underneath the redox peaks was integrated to measure the charge associated with the Ni redox, a well-established method to quantify the number of electrochemically accessible metal cations.[33] The inset of Figure 3a shows the measured charge in cyclic voltammograms recorded before and after exposing the sample to OER conditions (see methods for details). We compare this value to the charge expected for redox involving all Ni ions in a single atomic layer on the perovskite surfaces, which depends on the orientation. After the first OER cycles, the measured charge is similar to the expected charge of one Ni layer for (001). Yet for (111), the measured charge is twice as high as expected for a single layer even in the first cycle, indicating that more than one Ni layer participates in the redox phenomenon for the (111) facet, a point to which we will return below.

Methods: ... Quantification of charge during cyclic voltammetry was performed following refs. [33, 46]. The amount of charge was calculated using the EC lab software package by integration of the anodic peak after subtraction of a linear background (capacitive contribution).”

Comment: 4. In Figure 4b, I agree that the peak iii disappeared in sample 16h-CA. But I do not think the peak ii is new because it stays over there exactly same as the as-prepared and 1h CA samples. I hope an expert in XAS can give more comment on this issue.

Response: We agree that this point deserves a closer look. The peak (ii) is present in the pristine sample (Response Figure 6a) when only the perovskite phase is modelled. Its origin are the sidelobes of the peaks corresponding to mainly Ni-La (denoted as feature (iii) in Fig. 4c of the main text). Note that this spectral intensity does not correspond to any physical coordination shell in the material. An exemplary contribution is shown in Response Figure 6b, where we remove the hydroxide contribution from the fit of the sample after 16 h CA. As expected, there is still a contribution due to the sidelobes of the Ni-La peaks. But the overall fit does not satisfactorily describe the measured data. Even when we fit this data without allowing the characteristic hydroxide contribution, the data cannot be described satisfactorily, most severely seen in the intensity difference of peak (iii) in Response Figure 6c.

We added these considerations in the new Supporting Note 2 and included the figures for illustration and added a description to clarify that the nature of the peak changes.

While it is evident that the model applied is necessary to reproduce the data, we note that during consistency checks based on all additional experiments suggested by the referees, we noticed that the 16h CA XAS data previously reported in the main text corresponds to a severely degraded film (~2.5 V vs. RHE, previously incorrectly labelled with 1.75 V vs. RHE). Therefore, this dataset is now only used as a reference and we changed the description to “The latter represents an extreme condition, magnifying the expected changes and is used as a reference.” In response to our observation of the previous mislabelling, we added additional XAS analysis (difference spectra for XANES and EXAFS) for a sample which was CA-treated for 1h at 1.75V. These data confirm that our previous conclusions remain intact, even when taking this sample only as a severely degraded reference point: The surface under operating conditions is likely to form an oxyhydroxide-like phase based on its XAS signature. The structural integrity of the bulk is in line with the added TEM, XPS and XRD data. In addition, we shortened the XAS section and moved details to an extended Supplementary Note to improve readability.

Response Figure 6: Fourier transform of the EXAFS of the Ni-K edge collected on (111)-oriented LaNiO_3 samples. a) As prepared. There is spectral intensity near peak (ii). But it is almost fully explained by sidelobes of the Ni-La interactions (peak (iii)). Other contributions could be Ni-O interactions that were not modeled. Fit in Supplementary Table 1. (b, c) 16 h CA at 2.5 V vs. RHE. In (b), we removed the Ni-Ni interaction at 3.2 Å, which results in a worse fit than the one in Fig. 4c of the main text. No new parameter optimization was performed after removal of the Ni-Ni interaction from the fit. In (c), we performed parameter optimization (i.e., a new fit), still excluding the Ni-Ni interaction, fit results are shown in Supplementary Table 5. This fit is also worse than the one in Fig. 4c of the main text. These considerations indicate that the spectral intensity near 2.5 Å reduced distance are mainly due to sidelobes for the as prepared sample but due to a (hydr)oxide phase for the sample after 16 h CA at 2.5 V. Data are shown as circles and EXAFS fits as lines. Fourier transforms were performed between 35 and 550eV (above $E_0 = 8333$ eV) with a cosine window covering the first and last 10 % of the data range.

Changes to the text (changes highlighted in yellow): “The X-ray absorption near edge structure (XANES) and the extended X-ray absorption fine structure (EXAFS) of the Ni-K edge were collected on a pristine (111) $\text{LaNiO}_{3-\delta}$ film, and after chronoamperometry (CA) at 1.75 V vs RHE for 1 h (initial current density of 3.2 mA cm^{-2}) and after CA at 2.5 V vs RHE for 16 h (Figure 4a). The latter represents an extreme condition, magnifying the expected changes and is used as a reference.

...

The $\text{LaNiO}_{3-\delta}$ sample after 1 h CA is almost identical to the pristine one except for a minute shift of the rising part to lower energy, which may be a result of a similar transformation as observed after 16 h, albeit to a much smaller extent. As small changes on the surface are expected, we calculated differential XANES spectra ($\Delta\mu$ method). We plot the intensity difference between treated (1h/1.75V and 16h/2.5V) samples and as-grown sample in Fig. 4b. The qualitative similarity in the difference spectra supports that the changes clearly visible in the sample after 16 h at 2.5 V were also present on the surface of the sample after 1 h at 1.75 V. The quantitatively small differences observed for the sample after 1 h at 1.75 V are expected for a chemical transformation of the surface-near atomic layers, given the extended information depth of fluorescence-mode XAS.

... Moreover, the (111) sample operated for 16 h at 2.5 V exhibited no clear Ni-La peak (iii) and instead a new peak (ii) appeared, which is not expected for the perovskite structure. Similar observations were made previously for Co-based perovskites, where the peak was assigned to a metal-metal distance in edge-sharing octahedra, which is also supported by the similar position of peak (ii) to the edge-sharing reference LiNiO_2 (Fig. 4b). Thus, we assigned the new peak to edge-sharing Ni octahedra (Supplementary Fig. 4b,c) and note that the corner-sharing Ni octahedra, i.e., peak (iv), are resolved simultaneously. We again calculated differential spectra (ΔEXAFS), where we observe qualitatively similar features for both treated samples. For the sample after 1 h at 1.75 V, only small changes are expected because the transformation is confined to the surface. The difference spectra thus support that the changes visible after 16 h at 2.5 V were also present on the surface of the sample after 1 h at 1.75 V (Fig. 4d), again implying a hydroxide-like surface phase. ”

Comment: 5. In page 5, all Figure 3bs should be Figure 3a.

Thank you, we corrected the mistakes.

Reviewer #2 (Remarks to the Author):

Comment: In this work, F"ungerlings et al., studied the (001), (110) and (111) facet of $\text{LaNiO}_3\text{-}\delta$ electrocatalysts and revealed a facet- dependent activity by using combined electrochemical characterization and theoretical simulation approaches. They found that both (001) and (111) facets undergo a significant surface transformation into oxyhydroxide-like NiOO and the (111) facets obtain deeper transformation, which delivers best OER performance The author employed systematic DFT+U simulation to confirm that the lattice parameter of transformed layer matches better with the pristine (111) slabs, leading to the less lattice distortion and optimized binding energy of reaction intermediates. Actually, previously, the work regarding facet dependent OER activity of LaNiO_3 has been reported by same group (Frontier Chem, 2022, 10, 913419), and the major novelty of this work focuses on the facet dependent on-site phase transformation behavior including their dynamic protonation and lattice mismatch. I will not further consider the publication of this work until the author addresses following questions.

Response: We thank the referee for the overall very positive assessment and recommendation of our manuscript, and for identifying the key novelty of the present manuscript in contrast to our prior work. As the referee points out, the key novelty lies in the deep mechanistic insights into the active surface phase formation in different crystal orientations, which allows us to computationally predict the activity differences in different facets and understand the underlying mechanisms (which go beyond a simple match of the lattices). Our work in Frontier Chem, 2022, 10, 913419, on the other hand, is merely a perspective paper on general relationships of activity and stability in OER catalysts, in which we mention activity-stability data for different facets of perovskite oxides, LaNiO_3 being only one example, in a phenomenological compilation of electrochemical data to illustrate a general trend. The prior work did not contain a detailed discussion or elaboration the underlying mechanistic processes that we provide in the current manuscript.

We are happy to address the detailed questions below.

Comment: (1) First of all, the author claimed that (111) facet exhibits best activity and stability. However, the discussion about the relationship between facet dependent phase transformation and durability is poor. To be specific, I am curious about the worst stability and activity of (110) facet, as compared to (111) and (100) counterparts. Since (110) facet has been regarded as the one having least phase transformation, it intuitively indicates that this facet is more stable or electrocatalytic inert. Why does it decay so fast? What is the mechanism behind the difference in stability? The authors are required to clarify it by offering more convincing data and explanation.

Response: The focus of our study is the activity and the mechanistic insight into the origin of the activity differences. Since the (110)-facet is of intermediate activity and since the cyclic voltammetry data (Figure 3a) does not indicate a pronounced tendency for a surface transformation, we have concentrated on the (111) and (001) surface orientations, which represent the most and the least active orientation and which both show a clear tendency towards a surface transformation, in particular the former. For completeness and based on the referees' suggestion, we now included a detailed DFT analysis of the (110) surface.

Experimentally, differences in stability largely remain a phenomenological observation, but we emphasize that all facets exhibited sufficient durability for a careful and meaningful comparison of the activity, as demonstrated by the CP measurements in Fig 3e,f of the main text, where the (110) facet has almost identical performance over time and as a function of potential as the (100) facet. The observed stability is hence still an open question for all the facets. One may speculate that future experimental characterization pathways such as electrochemical STM as presented by the Zhang

group (<https://pubs.rsc.org/en/content/articlelanding/2023/SC/D2SC07034K>) may help in arriving at a more complete picture. We hope to contribute to the development of this field by supplying the additional characterization that the referees kindly requested. Following the suggestion of the referee, we have now included a detailed DFT+*U* analysis of the (110) surface. Accessing the stability of such a complex multicomponent system from a computational point of view is challenging and beyond the scope of the current manuscript. However, as a first step, we have determined the interface binding energy of the oxyhydroxide layer to differently terminated perovskite facets and find that the (111) system is energetically favored over (110) and (001). Regarding the overpotential, we find that a partially hydrogen-covered transformed (110) facet yields an overpotential of 0.48 V, slightly higher than 0.45 V for (111). For comparison, at the untransformed (110), the overpotential is considerably higher (0.77 V). Considering the experimental observation that the surface transformation is less pronounced for (110), this leads to a consistent trend between experiment and theory among the different facets.

Comment: (2) Since (001) and (111) could be terminated by NiO₂/LaO and Ni³⁺ /LaO₃, respectively. Do the author mean most of the (001) and (111) facets were terminated by NiO₂ and Ni³⁺, respectively? I am glad to see that the authors provide surface angle-resolved XPS to confirm the surface rich phase of Ni species. If so, why does the author still persist on constructing the models with LaO and LaO₃ termination which is literally impossible for (001) and (111) facets?

Response: We thank the referee for this question and are happy to supply additional computational data.

XPS: As we stated in the caption of SI Fig 3 of the previous submission, “Only minor changes in cation composition of the surface layer is observed.” We now add additional information on the surface vs. subsurface composition, obtained from angle dependent XPS measurements (resulting in different mean information depth $d=1.1$ nm and $d=2.1$ nm), as discussed in detail in our response to referee 1 and Response Figure 3 which supports the above observation.

DFT: Since the Ni cations in the transformed oxyhydroxide-like layers are expected to stem from the original perovskite catalyst, it is reasonable to assume that the subjacent LaNiO₃(111) surface is LaO₃-terminated. On the other hand, several Ni-containing perovskite layers are necessary to form the (laterally denser) oxyhydroxide-like layer. Thus, an interface with a Ni-terminated LaNiO₃(111) might be a viable option. Following the suggestion of the reviewer, we extended our DFT+*U* study to model NiOO(H) on Ni-terminated (111) LaNiO₃. As the updated Figure 5 (Response Figure 7) shows, the NiOO/Ni-LaNiO₃(111) and the NiOOH_{0.75}/LaO₃-LaNiO₃(111) systems compete in energy in the vicinity of the redox peak. Therefore, both were considered in the further OER analysis. In the updated Figure 6 (Response Figure 8), we now show the reaction free energies of OER-intermediates and the overpotentials for the oxyhydroxide-like layers on both the LaO₃- as well as the Ni-terminated LaNiO₃(111). The OER-overpotentials for both systems are very similar (0.45 V vs. 0.50 V).

Changes to the main text:

“In the transformed surfaces, the lateral density of Ni in the edge-sharing NiOO(H) layer is higher than in the perovskite layers. Consequently, more than one Ni-containing perovskite layer is involved in the formation of one layer of NiOO(H), raising the question which perovskite termination is present at the interface to the NiOO(H) layer. For the (001) facet, it was established that NiOO on a LaO-termination is more stable than on NiO₂.^[12] We compared the interface binding energy of NiOOH_{*x*}, where *x* denotes different amounts of hydrogen intercalated at the interface, to both possible terminations of the (110) and (111) facets, as well as the LaO-terminated (001) surface. As shown in Fig. 5, for the (111) oriented surfaces, the two surface structures, NiOOH_{0.75} on LaO₃-terminated

LaNiO₃(111) and NiOO on Ni-terminated LaNiO₃(111), compete in energy in the vicinity of the redox peak. Therefore, both were considered in the further analysis.”

Comment: (3) How does the author calculate, quantify or expect the amount the charge layer based on CV scan shown in Figure 3a? the detail calculation method should be given. Also, why does the author choose the fast scan rate of 500 mv/s ?

Response: We gladly provide further information on the observed and expected amount of charge. Quantification of charge during cyclic voltammetry: We followed the procedure discussed in detail in Shannon Boetcher's work, e.g. 10.1021/accountsr.1c00087, 10.1021/acs.chemmater.6b02796. The amount of charge was calculated using the EC lab software package by integration of the anodic peak after subtraction of a linear background (capacitive contribution). Comparison of peak areas for cathodic and anodic peaks led to negligible differences, as expected for a surface redox process (10.1021/acs.chemmater.6b02796). We chose the fast sweep rate of 500 mV/s from a dataset of 10, 30, 50, 70, 100, 150, 200, 250, 300, 350, 400, 450, 500 mV/s scans because of the highest absolute current values. Repeating the same analysis for different sweep rates from the same data set led to similar values within ~10% of the calculated charge.

Response Figure 9: Screenshot from the EC Lab software for the integration of the oxidation peak.

Expected amount of charge: We calculated the number of Ni atoms exposed on a perfectly flat surface using the geometric area exposed to the surface (0.441 cm^2), well justified by the negligible RMS roughness observed in AFM scans. Per orientation, we calculated the number of Ni atoms per area based on the 2D-projected unit cell, arriving at one Ni atom for 0.15, 0.22, 0.26 nm^2 for the (100), (110), (111) orientation.

We included this information in the methods section of the revised manuscript.

Changes to the text (changes highlighted in yellow): “The area underneath the redox peaks was integrated to measure the charge associated with the Ni redox, a well-established method to quantify the number of electrochemically accessible metal cations.[33] The inset of Figure 3a shows the measured charge in cyclic voltammograms recorded before and after exposing the sample to OER conditions (see methods for details).”

Methods: ... Quantification of charge during cyclic voltammetry was performed following refs. [33, 46]. The amount of charge was calculated using the EC lab software package by integration of the anodic peak after subtraction of a linear background (capacitive contribution).”

Comment: (4) There are some typos all through the manuscript, suggesting the carelessness of proofreading. For example, there is no inset in figure 3b. it should be in figure 3a. Only part of the EIS plot can be found in figure 3b. Detailed analysis and fitting of is lacking.

Response: Thank you for spotting the incorrect reference to the inset, we corrected it.

As for the EIS data, we intentionally show the zoom of the EIS data because we only use it to extract the uncompensated resistance R_u , i.e. the high frequency intercept with the real axis. We show the full data in Response Figure 10. But further analysis of this data at open circuit voltage is in our opinion not instructive, because no fast redox couple is present in solution.

Comment: (5) In Figure S4 of Pourbaix plot, the authors claim that at the untransformed, Ni-terminated surface of LaNiO₃(111), two coverages are close in energy under reaction conditions pH = 13 and $\phi = 1.68$ V versus RHE, either one monolayer (ML) *OOH or 0.75 ML *OH+0.25 ML *OOH. However, as long as we go back to Figure S4a, we can easily find that there is no unclear margin at this condition. Instead, the one monolayer OOH is predominant. I thought the author wished to talk about Figure S4b in which both scenarios are possible.

Response: We note that while in experiment, the voltage is measured vs RHE, in the DFT-based Pourbaix diagrams the voltage of the standard hydrogen electrode (SHE) is shown. Due to the pH-dependence when relating the former to the latter, the reaction conditions correspond to pH = 13 and $\phi = (1.68 - 0.0591 \cdot \text{pH}) \text{ V} \approx 0.91 \text{ V}$ vs SHE in Supplementary Figure S11a (S4a before revision), where the two coverages are of comparable stability. The axis label has been updated accordingly and the following sentence has been added to the caption of Supplementary Figures 11 (formerly 4) and the new Supplementary Figure 12:

Addition to Supplementary Figures 11, 12 captions:

“Note that the voltage in experiment is measured vs RHE, therefore the reaction conditions are found at $\phi = (1.68 - 0.0591 \cdot \text{pH}) = 0.91 \text{ V}$ vs SHE and pH = 13 in the diagram.”

...

“Note that the voltage in experiment is measured vs RHE, therefore the reaction conditions are found at $\phi = (1.72 - 0.0591 \cdot \text{pH}) = 0.91 \text{ V}$ vs SHE and pH = 13 in the diagram.”

Comment: (6) There have no TEM images to directly visualize the phase transformation, which are essentially required to confirm the reconstruction depth. In XANES plot, the spectra of samples after 16-hour anodic treatment are provided. Why is 16 h? Is 16 h long enough to realize the complete reconstruction? What if further extend the treatment time? To answer this set of questions, I would love to know the performance of the electrocatalyst after 16 h treatment first.

Response: We appreciate the referee's suggestion and performed additional experiments for up to 100 h of operation in response. However, we point out that the key message of our manuscript is not the long-term degradation behavior but the transformation occurring in the initial operation of LaNiO₃ electrocatalysts as a function of surface orientation.

The main implication of the additional experiments is: **The stability of the epitaxial thin films is fully sufficient to draw the conclusions presented in the manuscript.**

Response Figure 11: Chronoamperometry (CA) at 1.8 V vs. RHE of (111) LaNiO₃ for 100 h (without iR correction).

Electrochemical data: During the 100 h operation time (to our knowledge one of the longest activity tests reported for epitaxial thin film model systems), the activity of the (111) LaNiO₃ remains high: The film undergoes an initial transformation (with a small dip in activity followed by an activity enhancement by factor ~2). After a few hours, the activity remains comparably constant over the measurement time. Note that the CA data is not *iR*-corrected and that R_u is further increasing during the experiment. Note that the CA data is not *iR*-corrected and that R_u is further increasing during the experiment.

TEM: We performed HAADF STEM analysis of (001) and (111) LaNiO₃ samples before and after 100 h CA, to track the most severe changes during stability tests (Response Figure 12). We note that cross-sectional TEM relies on preparation of lamella using a focussed ion beam (FIB). While we used a protective carbon layer during lamella preparation, the top surface (~1-2 unit cells) may be damaged during FIB, implying that TEM is not suitable to test the hypothesis of *transformation* towards oxyhydroxide-like surface layers, which was the motivation for our focus on XAS in the original submission. Nevertheless, the TEM analysis provides valuable additional information regarding the *long-time degradation*: Even after 100 h of operation, the bulk of the film remained intact (in line with the XRD analysis provided in response to referee 1, detailed in Response Figure 5). Near the surface, we observe pit-like corrosion: The entire lamellas for both orientations were decorated with nm-sized amorphous regions with a depth of up to 3 nm. Within experimental resolution, the EDS composition of these regions and the surface is identical to the bulk of the film. These observations have important implications for stability-investigations of perovskite-type electrocatalysts and will form the basis for future studies focusing on the stability and deactivation mechanism rather than the initial transformation investigated in our manuscript.

We added the new information in Supplementary Note 3 and Supplementary Figure 8.

Response Figure 12: HAADF STEM analysis of LaNiO₃ samples after 100 h CA for (001) orientation (a, c) and (111) orientation (b,d). Pit-like corrosion is observed after this long-time stability tests (highlighted with red and yellow ellipses in panels a and b). These pits do not result from the FIB preparation; they are absent for lamella prepared from as-prepared (001) and (111) LaNiO₃ (*new Supplementary Fig. 2*). In addition, we observe extended defects in the bulk of the film, for samples both before and after CA (examples highlighted with red arrows). These were observed before (ref. 12 of the main text), connected to the chosen synthesis temperature, which was in turn necessary to obtain the desired Ni-termination in the as-prepared state.

While a dedicated environmental TEM (ETEM) study for LaNiO₃ is out of scope for the present work, we note that some of us recently used ETEM to demonstrate a dynamic rearrangement of the surface of La_{0.6}Sr_{0.4}CoO₃ epitaxial thin films (DOI 10.1021/jacs.2c07226, ref 17 of the main text). The results revealed the formation of a disordered phase of a few unit cells over time with applied potential, with is consistent with our experimental results for LaNiO₃ discussed in the present paper.

During consistency checks based on all additional experiments suggested by the referees, we noticed that the 16h CA XAS data previously reported in the main text corresponds to a severely degraded film (~2.5 V vs. RHE, previously incorrectly labelled with 1.75 V vs. RHE). Therefore, this dataset is now only used as a reference and we changed the description to “The latter represents an extreme condition, magnifying the expected changes and is used as a reference.” In response to our observation of the previous mislabelling, we added additional XAS analysis (difference spectra for XANES and EXAFS), which confirm that our previous conclusions remain intact, even when taking this sample only as a severely degraded reference point: The surface under operating conditions is likely to form an oxyhydroxide-like phase based on its XAS signature. The structural integrity of the bulk is

in line with the added TEM, XPS and XRD data. In addition, we shortened the XAS section and moved details to an extended Supplementary Note to improve readability.

Changes to the text (changes highlighted in yellow): “The X-ray absorption near edge structure (XANES) and the extended X-ray absorption fine structure (EXAFS) of the Ni-K edge were collected on a pristine (111) $\text{LaNiO}_{3-\delta}$ film, and after chronoamperometry (CA) at 1.75 V vs RHE for 1 h (initial current density of 3.2 mA cm^{-2}) and after CA at 2.5 V vs RHE for 16 h (Figure 4a). The latter represents an extreme condition, magnifying the expected changes and is used as a reference.

...

The $\text{LaNiO}_{3-\delta}$ sample after 1 h CA is almost identical to the pristine one except for a minute shift of the rising part to lower energy, which may be a result of a similar transformation as observed after 16 h, albeit to a much smaller extent. As small changes on the surface are expected, we calculated differential XANES spectra ($\Delta\mu$ method). We plot the intensity difference between treated (1h/1.75V and 16h/2.5V) samples and as-grown sample in Fig. 4b. The qualitative similarity in the difference spectra supports that the changes clearly visible in the sample after 16 h at 2.5 V were also present on the surface of the sample after 1 h at 1.75 V. The quantitatively small differences observed for the sample after 1 h at 1.75 V are expected for a chemical transformation of the surface-near atomic layers, given the extended information depth of fluorescence-mode XAS.

...

Moreover, the (111) sample operated for 16 h at 2.5 V exhibited no clear Ni-La peak (iii) and instead a new peak (ii) appeared, which is not expected for the perovskite structure. Similar observations were made previously for Co-based perovskites, where the peak was assigned to a metal-metal distance in edge-sharing octahedra, which is also supported by the similar position of peak (ii) to the edge-sharing reference LiNiO_2 (Fig. 4b). Thus, we assigned the new peak to edge-sharing Ni octahedra (Supplementary Fig. 4b,c) and note that the corner-sharing Ni octahedra, i.e., peak (iv), are resolved simultaneously. We again calculated differential spectra (ΔEXAFS), where we observe qualitatively similar features for both treated samples. For the sample after 1 h at 1.75 V, only small changes are expected because the transformation is confined to the surface. The difference spectra thus support that the changes visible after 16 h at 2.5 V were also present on the surface of the sample after 1 h at 1.75 V (Fig. 4d), again implying a hydroxide-like surface phase.”

Comment: (7) Usually, hard X-Ray XAS is a bulk sensitive characterization tool. Based on the EXAFS data, I can expect the deep reconstruction of (111) facet. However, the spectra for (100) and (110) are missing, which are required to show for fair comparison. Moreover, a table of fitting parameter should be given.

Response: We agree that our XAS measurements are bulk-sensitive, since we measured in fluorescence mode. We mention again that the deep reconstruction of the (111) facet observed in the previous submission is a result of severe degradation. All data presented above indicate that only a small surface layer is affected by electrochemical treatment under regular OER potentials for up to 100 h. This is in line with the observations for the (111) facet after 1 h operation at 1.75 V vs RHE in the main text. Our EXAFS analysis is focused on the (111) surface and we prefer to keep this focus. Yet, we now included the measurements on the (001) surface to address the reviewer’s comment and included it in the Supplementary Figure 6, reproduced here as Response Figure 13. No samples with (110) were measured due to the scarcity of beamtime. Additional beamtime is currently impossible due to the Cyber attack on the Helmholtz Center Berlin and Bessy II synchrotron.

Response Figure 13: Fourier transform of the EXAFS of the Ni-K edge collected on (001)-oriented $\text{LaNiO}_{3-\delta}$ samples.

We added to the main text: “For the (001) surface after 1 h CA at 1.75 V vs. RHE, the EXAFS remains similar to the pristine state, akin to the (111) surface (Supplementary Fig. 6). We note that EXAFS peaks indicate the bond distances of highest occurrence. The data thus indicate that the films mostly remained in perovskite phase, in line with our XRD, XPS and STEM characterization (Supplementary Figs. 7-9). In Supplementary Note 3, we extensively discuss how the additional structural characterization confirms that only a thin surface layer transforms during operation at 1.75 V vs RHE.”

The fitting parameters were given in Supplementary Tables 1-4 of the original submission. We made the reference to the tables more prominent, added a reference to the caption of Fig. 4 in the revision and added two new table for the (001) facet.

Changes to the main text: “A detailed discussion of the EXAFS fits and the extracted interatomic distance, R , is provided in Supplementary Note 2, Supplementary Fig. 5 and Supplementary Tables 1-7”

Comment: (8) Basically, the theoretical simulation part of this work is well-organized. I just have one more question regarding the calculation of reaction Gibbs free energy. As the author claims: At the transformed oxygen-terminated LaNiO_3 surfaces, the OER cannot take place on top of cation sites like on the perovskite surfaces, but is performed via the lattice oxygen mechanism at a surface oxygen site.” Usually, the absorbent evolution mechanism happens on metal site only with the reaction intermediate of OH, O and OOH. However, for lattice oxygen mechanism, the reaction coordinate is different. Nevertheless, it seems like the intermediate for AEM is directly used for LOM, as shown in Figure 6. Please explain it.

Response: We thank the referee for pointing this out. There are different types of lattice oxygen mechanisms discussed in the literature (<https://doi.org/10.1039/D1EE01277K>, new reference 41). As there are no exposed cation sites on the oxyhydroxide(-like) surface layer studied in this work, the oxygen-vacancy-site mechanism (OVSM) is the only possible LOM that can take place. This type of LOM involves similar intermediates to the AEM with the distinction that the OER starts in a lattice oxygen vacancy. We have clarified this point in the updated manuscript:

We added to the main text: “Since the transformed NiOO(H) surfaces are oxygen terminated (with each surface oxygen being coordinated by three Ni atoms), the OER can take place only via the oxygen vacancy site mechanism (OVSM). This mechanism belongs to the lattice oxygen mechanisms (LOM), but consists of the same reaction intermediates as the adsorbate evolution mechanism (AEM), starting at an oxygen vacancy in the surface layer (denoted * in this work), with the *O intermediate being in this case part of the lattice.[41]”

Reviewer #3 (Remarks to the Author):

Comment: The manuscript reported by Pentcheva et al. investigated the crystal-facet-dependent surface transformation and contribution to the OER activity of LaNiO_{3-δ} from both the perspective of the experiment and DFT+U calculation. The results demonstrated that the (111) facet possesses a lower overpotential than both (001) and (110) facets. The reasons can be ascribed to that the transformed surface is thicker for (111) than for (001) and exhibits a protonated interface, as well as a better match between the transformed surface and the underlying perovskite. This leads to a favorable binding energy of intermediates.

Overall, this work is well-designed and novelty. But a major revision is recommended before considering publication in Nature Communication.

Response: We thank the referee for the overall very positive assessment and recommendation of our manuscript. We are happy to address the detailed questions below.

Detailed suggestions are as follows:

Comment: (1) The structure and content of the manuscript need to be revised and simplified, especially in DFT section. Only the most relevant results should be present in the maintext, because the current analysis and expression are ambiguous.

Response: In response to the referee comments, we removed part of the detailed XAS analysis from the main text and included it in Supplementary Note 2, instead. The additional experiments the referees kindly suggested are mostly covered in the new Supplementary Notes, too, to keep the main text as concise as possible.

The DFT section was rewritten to be more concise and readable. It now concentrates on the most important aspects: (i) we compare the thermodynamic stability of NiOO(H) on the different perovskite orientations and terminations and establish the most stable interfaces that are then used as model systems for further calculations. (ii) we establish the most stable coverage of both untransformed and transformed surfaces under reaction conditions in a Pourbaix diagram which is shown in the Supplementary Material. (iii) we present the overpotentials for the untransformed and transformed systems and in particular point out the higher overpotential and the different potential-determining step for the transformed (001) facet. (iv) we present a detailed analysis of the structural and electronic properties and identify a crucial structural differences of the NiOO(H) layers at the three facets, namely a strong distortion of NiOOH at (001) and (110) due to lattice mismatch and a much better fit for the (111) orientation. We also ascertain significant differences and variation of the ratio of Ni²⁺ and Ni³⁺ to Ni⁴⁺ cations in the surface layer during OER which affects the binding of intermediates and determines the trends in the OER overpotential.

Comment: (2) (001), (110), and (111) facets are experimented, why only both (001) and (111) facets are considered in DFT calculation? The DFT calculation based on (110) facet needs to be supplemented. Because these results are essential to confirm the consistent conclusion with the experiment that (111) is favorable to (001) and (110) facets.

Response: Our initial focus was to consider the most and the least active surface orientation. Motivated by the referee's suggestion, we have now extended our DFT study to model the transformed and untransformed (110)-oriented LaNiO_3 surfaces, and included additional data and discussion in the main text and SI of the revised manuscript.

The stability under reaction conditions is shown in in the SI Figure 4. The overpotential for the transformed partially H-covered $\text{LaNiO}_3(110)$ surface (0.48 V) is slightly higher than that of the transformed $\text{LaNiO}_3(111)$ (0.45 V). In experiment differences in the redox peak area for the different facets suggest that the (110) facet shows the lowest tendency towards transformation. Hence, most likely the actual overpotential is the one obtained for an untransformed (110) surface, (0.77 V). Overall we trend of OER activity for the three surface orientations from the DFT+U simulations is consistent with the experimental observations and provides also insight into the origin of these trends with the detailed analysis of the structural and electronic properties described above. In particular, as shown in Figure 7/Response Figure 14 the $\text{NiOO}(\text{H})$ at the (110) is also strongly distorted due to the mismatch to the underlying perovskite lattice, analogous to the (001) facet, but the bonding to the underlying perovskite is much different which together with differences in the oxidation states of the surface Ni sites affects the energetics.

Comment: (3) In Fig. 3b, why about 20 Ω uncompensated resistance difference between (001) and (111) facets are tested in Nyquist impedance plots?

The facets maybe not dictate such huge solution resistance differences.

Response: Thank you for pointing this out. Indeed, the variations in uncompensated resistance are not related to the facet. Considering an arbitrary set of three samples per orientation, we find resistances from 50-71.5 Ω for (001) and 55-75 Ω for (111). These variations are typical for the used cell geometries, where small differences in electrode placement may occur. An overview comparison using similar cells and similar epitaxial thin films indicated an R_u confidence interval of $\pm 16 \Omega$ (10.1021/acs.jpcc.6b07654). More importantly, the activity difference are also observed without iR_u - correction (although (111) had higher resistance values than (001), Response Figure 15). We now included a note that the differences in uncompensated resistance are related to small changes in placement of the reference electrode in the cell, rather than a result of the samples.

Changes to the text: see next comment.

Response Figure 15: Exemplary cyclic voltammetry in the OER potential regime, showing the average between anodic and cathodic sweep of the second cycle, with (dashed lines) and without (solid lines) iR_u -correction.

Comment: (4) The applied iR -correction rate should be highlighted for its huge effect on OER results. The relevant reports (Energy Environ. Sci. 11 (2018) 744-771; Materials Today Energy 32 (2023) 101246) have highlighted this point.

Response: Thank you for pointing out these interesting references. While we agree how crucial the iR -correction procedure is for reporting “high-activity” catalysts, we emphasize that we use a comparison between model systems, applying an identical procedure for iR correction. As discussed above (comment 3), the iR correction is not responsible for the observed activity trends. We cite one of the suggested works in the revised methods section.

To clarify this, we have made the following **changes to the text**: “ R_u was extracted from the intercept of the high-frequency impedance data with the $Re(Z)$ axis using linear extrapolation. Differences in uncompensated resistance are related to small changes in placement of the reference electrode in the cell, rather than resistivity differences across different orientations. We note that for reporting absolute values of the OER activity, the iR_u correction rate and procedure may have a large effect [45]. But we emphasize that our conclusions rely on one-to-one comparison of model systems, where the same activity trends are observed with and without iR_u correction.”

Comment: (5) A thicker oxyhydroxide-like NiOO surface transformation in (111) facet may increase the ECSA. Thus, ECSA test can consider added.

Response: Thank you for the suggestion. We included the linear fits of our double layer capacitance measurements in response. Both before and after OER, the (001) and (111) facets differ by less than ~1.1% in ECSA. The observed ~10 % roughening during OER is expected, partially due to formation of a new surface phase. We included this information in the revised text.

Response Figure 16: Double layer capacitance measurements for (111) and (001) samples, indicating only ~1% difference in exposed area, both before and after OER. Data points represent average cathodic and anodic currents during CV in the double layer region (0.96 to 1.1 V vs. RHE) and lines represent proportional fits. The capacitance extracted from the slope is mentioned in each panel.

Changes to the text: “Because of the low roughness, we approximate the actual oxide surface area with the geometric area. Double layer capacitance measurements of the different facets confirm differences within 1.1 %.”

Comment: (6) A further long-term galvanostatic hold (such as ≥ 100 h) would induce a full of surface oxyhydroxide transformation, at that moment, will the similar conclusion that (111) is favorable to

(001) and (110) be obtained?

Response: Thank you for the suggestion and hypothesis. In response, we performed additional experiments for up to 100 h of operation in response, and found consistent results with our previous submission. However, we point out that the key message of our manuscript is not the long-term degradation behavior but the transformation occurring in the initial operation of LaNiO_3 electrocatalysts as a function of surface orientation.

We observe that even after 100h of operation, the bulk of the film stays intact and degradation is mainly limited to the surface. Our observations imply that only the surface undergoes transformation (and degradation starts at the surface). While deeper interpretation of the degradation behavior is beyond the scope, we note that the data shows that under all experimental conditions the samples stay sufficiently stable to allow conclusions on the surface phase transformation, which is the main point of our present manuscript. Accordingly, the (111) facet stays the most active over time. In the revised manuscript, we added additional TEM, XPS, and XRD analysis (Supplementary Note 2, Supplementary Figures 7-9) and refined the XAS analysis. For convenience, we replicate the data below.

Response Figure 17: Chronoamperometry (CA) at 1.8 V vs. RHE of (111) LaNiO_3 for 100 h. We note that our long-time stability tests of comparable samples always showed that the (111) sample stayed more active than the (001) counterparts.

Electrochemical data: During the 100 h operation time (to our knowledge one of the longest activity tests reported for epitaxial thin film model systems), the activity of the (111) LaNiO_3 remains high: The film undergoes an initial transformation (with a small dip in activity followed by an activity enhancement by factor ~ 2). After a few hours, the activity remains comparably constant over the measurement time. Note that the CA data is not iR -corrected and that R_u is further increasing during the experiment.

XPS: As we stated in the caption of SI Fig 3 of the previous submission, “Only minor changes in cation composition of the surface layer is observed.” We now add additional information on the surface vs. subsurface composition, obtained from angle dependent XPS measurements (resulting in different mean information depth $d=1.1$ nm and $d=2.1$ nm). The overall Ni/La ratio at $d=1.1$ nm is higher than for $d=2.1$ nm for all samples, indicating predominant Ni termination in the as-prepared state for both (100) and (111) orientations. For operation up to 16h, the Ni/La ratio remains constant, as shown exemplarily for the (111) orientation in Response Figure 18. This indicates that on average, the transformed surface is as Ni-rich as the predominantly Ni-terminated as-prepared state.

Only after additional stability testing up to a total operation time of 100 h, we find that for both (100) and (111) samples, the Ni/La changes: While the surface remains slightly Ni-rich, the overall cation ratio approaches 1:1. But we note that the XPS signal intensity decreased by a factor of roughly 10 from the pristine state to the 100h CA state, indicative of an increased surface roughening (in line with the XRR and TEM results discussed below), making Ni/La ratios extracted from the area-integrated technique less reliable.

Response Figure 18: a) Cation ratio from XPS for a (111) LaNiO₃ sample as a function of CA time (CA interrupted for intermediate XPS measurements). b) Cation ratio from XPS for a (001) LaNiO₃ sample, with a continuous CA measurement for 100 h. Different probing depths d are obtained with different photoemission angles (see manuscript).

TEM: We performed HAADF STEM analysis of (001) and (111) LaNiO₃ samples before and after 100 h CA, to track the most severe changes during stability tests (Response Figure 19). We note that cross-sectional TEM relies on preparation of lamella using a focussed ion beam (FIB). While we used a protective carbon layer during lamella preparation, the top surface (~1-2 unit cells) may be damaged during FIB, implying that TEM is not suitable to test the hypothesis of oxyhydroxide-like surface layers, which was the motivation for our focus on XAS in the original submission. Nevertheless, the TEM analysis provides valuable additional information: Even after 100 h of operation, the bulk of the film remained intact (in line with the XRD analysis below). Near the surface, we observe pit-like corrosion: The entire lamellas for both orientations were decorated with nm-sized amorphous regions with a depth of up to 3 nm. Within experimental resolution, the EDS composition of these regions and the surface is identical to the bulk of the film. These observations have important implications for stability-investigations of perovskite-type electrocatalysts and will form the basis for future studies focusing on the stability and deactivation mechanism rather than the initial transformation investigated in our manuscript.

Response Figure 19: HAADF STEM analysis of LaNiO₃ samples after 100 h CA for (001) orientation (a, c) and (111) orientation (b,d). Pit-like corrosion is observed after this long-time stability tests (highlighted with red and yellow ellipses in panels a and b). These pits do not result from the FIB preparation; they are absent for lamella prepared from as-prepared (001) and (111) LaNiO₃ (new Supplementary Fig. 2). In addition, we observe extended defects in the bulk of the film, for samples both before and after CA (examples highlighted with red arrows). These were observed before (ref. 12 of the main text), connected to the chosen synthesis temperature, which was in turn necessary to obtain the desired Ni-termination in the as-prepared state.

XRD/XRR: We performed X-ray reflectivity (XRR) and X-ray diffraction (XRD) before and after 100 h CA (See Response Figure 20). Overall, the data confirm that the bulk of the films did not change to a large extent, in line with the TEM analysis. The XRR data is consistent with an increased surface roughening and a similar thickness for all film (with a ~1 μc thicker (111) film after 100 h). The nm-sized amorphous regions observed in TEM make the extraction of quantitative figures from XRR difficult. The (bulk-sensitive) XRD data indicates a similarly high crystallinity (similarly pronounced Laue fringes) and similar lattice constant before and after the stability test, with small thickness variations (variations in the period of the Laue fringes).

Response Figure 20: XRR and XRD analysis before and after 100 h CA for (001) orientation (a, c) and (111) orientation (b,d).

Conclusion: We observe local near-surface amorphization after extended stability tests, but otherwise the LaNiO_3 samples remain highly-ordered model systems in the bulk. We conclude that the long-time stability observed for the bulk of the films (XRD, TEM) and the comparably stable surface cation ratio (for at least 16 h of operation) support the hypotheses in our manuscript. We added this information to the new Supplementary Note 3. All of these observations are in line with the similarity of the XANES and EXAFS signatures of the (111)-oriented films after 1h operation (Fig. 4 in the main text).

We trust that with the additional measurements, calculations and comprehensive analysis we could resolve the concerns of the referees and that the substantially rewritten manuscript presents a clearly improved contribution of substantial interest to the diverse Nature Communications readership.

REVIEWERS' COMMENTS

Reviewer #1 (Remarks to the Author):

I am pleased with the authors' responses and revisions and recommend accepting their submission for publication in Nature Communications.

Reviewer #2 (Remarks to the Author):

I think the authors have completely addressed my concerns raised in the first round. I recommend the publication of this work.

Reviewer #3 (Remarks to the Author):

The authors have made substantial changes in the revised manuscript. Most importantly, the manuscript now has a good discussion of the results. I thank the authors for providing additional data to improve the manuscript and the addition of additional data to discuss stability and iR-correction is also greatly appreciated. I would like to recommend its publication in current form now.